# Metabolic and transcriptomic reprogramming during contact inhibition-induced quiescence is mediated by YAP-dependent and YAP-independent mechanisms

Soeun Kang [1], Maciek R. Antoniewicz[2] & Nissim Hay [1,3] ✉

Metabolic rewiring during the proliferation-to-quiescence transition is poorly understood. Here, using a model of contact inhibition-induced quiescence, we conducted [13]C-metabolic flux analysis in proliferating (P) and quiescent (Q) mouse embryonic fibroblasts (MEFs) to investigate this process. Q cells exhibit reduced glycolysis but increased TCA cycle flux and mitochondrial respiration. Reduced glycolytic flux in Q cells correlates with reduced glycolytic enzyme expression mediated by yes-associated protein (YAP) inhibition. The increased TCA cycle activity and respiration in Q cells is mediated by induced mitochondrial pyruvate carrier (MPC) expression, rendering them vulnerable to MPC inhibition. The malate-to-pyruvate flux, which generates NADPH, is markedly reduced by modulating malic enzyme 1 (ME1) dimerization in Q cells. Conversely, the malate dehydrogenase 1 (MDH1)-mediated oxaloacetate-to-malate flux is reversed and elevated in Q cells, driven by high mitochondrial-derived malate levels, reduced cytosolic oxaloacetate, elevated MDH1 levels, and a high cytoplasmic NAD$^+$/NADH ratio. Transcriptomic analysis revealed large number of genes are induced in Q cells, many of which are associated with the extracellular matrix (ECM), while YAP-dependent and cell cycle-related genes are repressed. The results suggest that high TCA cycle flux and respiration in Q cells are required to generate ATP and amino acids to maintain *de-novo* ECM protein synthesis and secretion.

Glucose, a major nutrient of most mammalian cells, is metabolized in glycolysis, the pentose phosphate pathway (PPP), the hexosamine pathway, and subsequently in the tricarboxylic acid (TCA) cycle. Glycolysis converts glucose into precursor metabolites for amino acid and fatty acid synthesis and generates adenosine triphosphate (ATP) and nicotinamide adenine dinucleotide (NADH). In glycolysis, 2 ATP and 2 NAD$^+$ are consumed, with the production of 4 ATP, 2 NADH, and 2 pyruvates per molecule of glucose. As glycolysis requires NAD$^+$, the recycling of NADH back into NAD$^+$ is pivotal for optimum glycolytic flux[1]. The PPP utilizes glucose for the production of nicotinamide adenine

[1]Department of Biochemistry and Molecular Genetics, College of Medicine, University of Illinois at Chicago, Chicago, IL, USA. [2]Chemical Engineering Department, University of Michigan, Ann Arbor, MI, USA. [3]Research and Development Section, Jesse Brown VA Medical Center, Chicago, IL, USA. ✉e-mail: nhay@uic.edu

dinucleotide phosphate (NADPH) and ribose 5-phosphate (R5P), which are required for redox homeostasis and fatty acid and nucleotide synthesis, respectively[2]. Pyruvate is further oxidized in the TCA cycle and generates additional NADH and flavin adenine dinucleotide (FADH$_2$), which in turn generate more ATP in the electron transport chain (ETC) in mitochondria. Mitochondrial NADH is oxidized upon donating its electrons to complex I (NADH: ubiquinone oxidoreductase) of the ETC, and mitochondrial NAD$^+$ is generated[3]. As both glycolysis and the TCA cycle are oxidative pathways, the oxidized form, NAD$^+$, serves as a cofactor for enzymes involved in these pathways, and NADH is produced[4]. Then, NADH produced by the TCA cycle links it to cellular energy generation, carrying electrons to the ETC on the inner mitochondrial membrane[5]. As the TCA cycle and ETC require NAD$^+$ and NADH, respectively, an optimal NAD$^+$/NADH ratio is needed for efficient mitochondrial metabolism.

Fibroblasts, the most common cell types in mammalian connective tissue, proliferate in the embryonic state, but with age, they arrest in the G$_1$ phase of the cell cycle[6]. Therefore, healthy fibroblasts in normal tissue are mostly in a quiescent state, representing a reversible exit from the cell cycle, and they can re-enter the cell cycle upon special circumstances such as wounding[7,8]. To supplement ATP and building blocks for macromolecule synthesis during the transition to the proliferative state, proliferating cells undergo metabolic reprogramming, such as shifting their metabolism to more anabolic pathways with an increased nutrient uptake and glycolysis[9–11]. It was previously reported that contact-inhibited quiescent human fibroblasts have reduced glycolysis but are still metabolically active[12]. However, how the metabolic flux is reprogrammed and regulated in quiescent cells is not fully understood. Therefore, we conducted quantitative metabolomics and $^{13}$C-metabolic flux analysis ($^{13}$C-MFA)[13] to characterize the metabolic differences and transcriptomic analysis between proliferating (P) and quiescent (Q) mouse embryonic fibroblasts (MEFs). We observed changes in glucose consumption, glycolysis, the TCA cycle, malic enzyme 1 (ME1)-mediated flux, and malate dehydrogenase 1 (MDH1)-mediated flux during the transition from a proliferative to a quiescent state. The decrease in glycolytic fluxes in Q cells is mediated by the suppression of yes-associated protein (YAP) activity, which is regulated by E-cadherin-mediated cell–cell interactions. Transcriptomic analysis confirmed that YAP target genes are downregulated in Q cells and restored by E-cadherin deletion. In contrast, the TCA cycle and mitochondrial metabolism are markedly elevated in Q cells, and pyruvate is diverted to the mitochondria instead of to lactate production. The diversion of pyruvate to the mitochondria is mediated by elevated expression of mitochondrial pyruvate carriers (MPC1 and MPC2), and reduced NAD$^+$ demand. ME1-mediated flux is markedly reduced in Q cells by modulating ME1 dimerization. The MDH1-mediated flux is reversed and elevated in Q cells, which is likely mediated by the high level of mitochondrial-derived malate, reduced cytosolic oxaloacetate, high level of MDH1, and relatively high cytoplasmic NAD$^+$/NADH ratio. The flux analysis was done mainly in MEFs, but we found similar trends such as reduced glycolysis and increased OXPHOS in other mouse and human fibroblasts as well as in epithelial cells. We used contact inhibition in the presence of growth factors to induce quiescence. However, when quiescence was induced by serum deprivation, which is governed by the lack of growth factors both glycolysis and OXPHOS were reduced.

Finally, many genes are induced in Q cells, including genes associated with the extracellular matrix (ECM). Thus, Q cells display massive reprogramming both metabolically and transcriptionally. The elevated TCA cycle flux and respiration in Q cells could be required to maintain ECM protein synthesis and secretion, which demands high ATP and amino acids generated by mitochondria, and particularly proline, which is enriched in the collagen family of proteins[14]. As the main roles of fibroblasts is to produce the ECM, this underscores the necessity for active mitochondrial metabolism to effectively support their essential cellular functions.

## Results

### Glycolytic flux is decreased, while TCA cycle flux and respiration are markedly increased in quiescent cells

To determine the metabolic changes occurring during the transition from a proliferative to a quiescent state, we generated MEFs immortalized with dominant-negative p53 as previously described[15,16]. The cells were analyzed while actively proliferating (2 days after plating) and when they were quiescent (contact-inhibited for 6 days, 11 days after plating) (Fig. 1a). In Q cells, the levels of p27$^{Kip1}$, a cyclin-dependent kinase inhibitor, were significantly increased, and concomitantly, the cell cycle distribution was mostly in G$_1$ phase with significantly reduced S phase and G$_2$/M phase (Fig. 1a, b). Transcriptomic analysis in P versus Q cells surprisingly showed that more genes were upregulated in Q cells; among 13,022 significantly differentially expressed genes, 7131 genes were upregulated, and 5891 genes were downregulated in Q cells (Supplementary Data 1). The heatmap, using stricter filtering criteria ($q < 0.01$ and logFC $< -2$ or logFC $> 2$), shows that out of 3319 genes, 2478 were upregulated and 841 were downregulated in Q cells. Gene set enrichment analysis (GSEA) was used to understand the pathways that were differentially regulated in P versus Q cells. Specifically, Q cells in comparison to P cells showed significantly downregulated gene sets associated with the cell cycle checkpoint (Supplementary Fig. 1b–d). These results show that P cells and Q cells have distinct gene expression profiles.

As expected, P cells exhibited higher glycolytic activity than Q cells, as confirmed by the extracellular acidification rate (ECAR) (Fig. 1c, d). To quantify metabolic fluxes in both P and Q cells, we employed $^{13}$C-metabolic flux analysis ($^{13}$C-MFA)[13,17–20] with a parallel labeling strategy using two isotopic tracers, namely, [1,2-$^{13}$C$_2$]-glucose and [U-$^{13}$C] glutamine, as previously described[21]. The flux analysis (Fig. 1e, Supplementary Fig. 2a, b, and Supplementary Data 2 and 3) showed that glucose uptake was modestly reduced, and glycolysis was reduced by approximately 2-fold in Q cells. Consistent with the flux analysis, [1,2-$^{13}$C$_2$]-glucose tracing experiments showed a significantly lower abundance of (M + 2) glycolytic metabolites, such as G6P, DHAP, pyruvate, and lactate, in Q cells than in P cells (Supplementary Fig. 2c, d). Although glycolytic flux was reduced in Q cells, no change in PPP flux was observed (Fig. 1e), which was also reflected by measuring the ratio of DHAP (M + 1) derived from the PPP versus DHAP (M + 2) derived from glycolysis with [1,2-$^{13}$C$_2$]-glucose tracing (Fig. 1f).

The flux of pyruvate to lactate was markedly decreased in Q cells (Fig. 1e and Supplementary Fig. 2b), which was also reflected by a 4-fold decrease in lactate excretion (Fig. 1d). Instead, there was a dramatic increase in pyruvate transport to the mitochondria (Fig. 1e and Supplementary Fig. 3a). Consequently, there was a dramatic increase in all TCA cycle reactions in Q cells (Fig. 1e and Supplementary Fig. 3a). Interestingly, the conversion of pyruvate to acetyl-CoA was markedly increased in Q cells, but the conversion of pyruvate to oxaloacetate is increased even more reaching to about 15-fold higher than in P cells (Fig. 1e). Consistent with high flux of TCA cycle, the mitochondrial membrane potential ($\Delta\Psi_m$) and basal respiration measured by the oxygen consumption rate (OCR) were higher in Q cells than in P cells (Fig. 1g–j). The higher respiration is likely not due to an increase in the number of mitochondria, but we found that mitochondria are more elongated in Q cells (Supplementary Fig. 3b, c), which could indicate fused and more active mitochondria[22–24]. This finding suggests that Q cells produce ATP preferentially from mitochondrial respiration. Finally, the flux of malate to pyruvate in mitochondria catalyzed by malic enzyme 2 (ME2) and the transport of malate from mitochondria to the cytosol are substantially induced in Q cells. Additionally, in Q cells, cytoplasmic malate is not converted to pyruvate catalyzed by ME1, generating NADPH, but rather, is converted to oxaloacetate (OAC), which is in turn converted to phosphoenolpyruvate (PEP) which concomitantly generate more pyruvate (Fig. 1e).

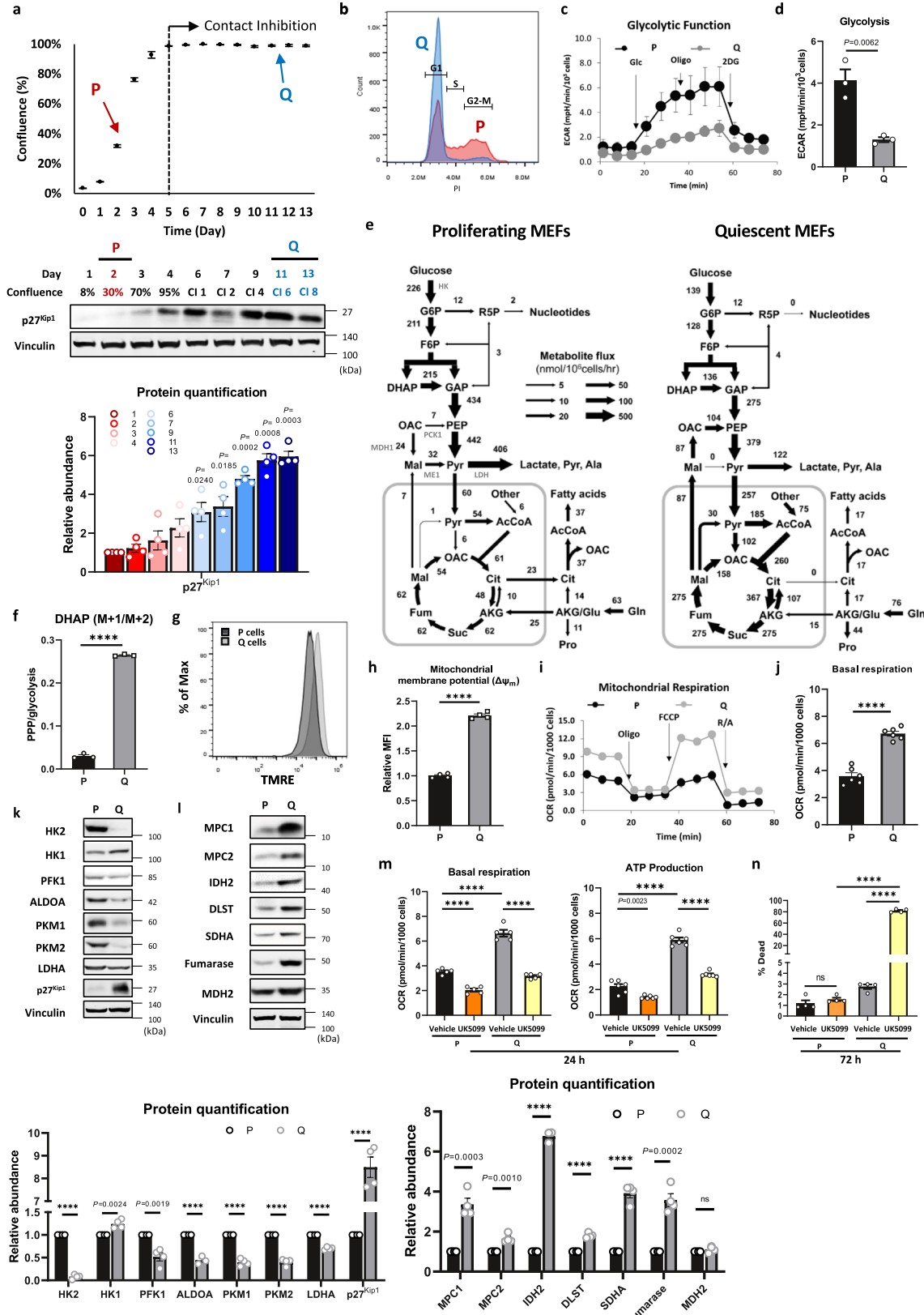

As the glycolytic flux rate is controlled by glycolytic enzyme expression[25], the expression levels of glycolytic enzymes in P versus Q cells were assessed (Fig. 1k). The expression of most enzymes in glycolysis was significantly downregulated in Q cells compared with P cells. In particular, hexokinase 2 (HK2), phosphofructokinase 1 (PFK1), and pyruvate kinase M1-2 (PKM1-2), the rate-limiting enzymes of glycolysis, were significantly downregulated, although p27[Kip1] levels were significantly increased in Q cells at both the protein and mRNA levels (Fig. 1k and Supplementary Fig. 2e). GSEA also showed that the expression of gene sets associated with glycolysis was significantly reduced in Q cells (Supplementary Fig. 2f, g).

**Fig. 1 | Glycolytic flux is decreased, while TCA cycle flux and respiration are markedly increased in quiescent cells. a** Growth curve and immunoblots with protein quantification of p27$^{Kip1}$ in MEFs. Proliferating cells (P): 2 days after culture; quiescent cells (Q): contact-inhibited for 6 days (11 days after plating). Values are the mean ± SEM of four independent experiments. *P* values by unpaired two-tailed student's *t*-test are indicated. Representative immunoblot is shown. **b** Representative histogram of the DNA content of P and Q MEFs after PI staining. The histogram is representative of three independent experiments. G1, S, and G2-M phases are indicated. **c, d** Extracellular acidification rate (ECAR) and individual glycolysis parameters of P and Q MEFs. Data are presented as the mean ± SEM of three biologically independent samples. *P* value by unpaired two-tailed student's *t*-test is indicated. Glc glucose, Oligo oligomycin, 2DG 2-deoxyglucose. **e** Metabolic flux maps for P and Q MEF cells. The flux maps were determined using $^{13}$C-MFA of multiple datasets at isotopic steady state as described in Methods. **f** Relative PPP/glycolysis represented by the ratio of intracellular 1x$^{13}$C DHAP (M + 1) over 2x$^{13}$C DHAP (M + 2) after 25 mM [1,2-$^{13}$C$_2$]-glucose labeling for 24 h. Means ± SEMs (*n* = 3) are shown. ****P* < 0.0001 using unpaired two-tailed *t*-test. **g, h** Mitochondrial membrane potential (ΔΨ$_m$), as reflected by the TMRE fluorescence, of P and Q MEFs. The mean fluorescence intensity (MFI) was calculated based on four independent experiments. Values are the mean ± SEM (*n* = 4). ****P* < 0.0001 using unpaired two-tailed *t*-test. **i, j** Oxygen consumption rate (OCR) and individual parameters for basal respiration of P and Q MEFs. Values are the mean ± SEM of six biologically independent experiments. ****P* < 0.0001 using unpaired two-tailed *t*-test. ****P* < 0.0001 using unpaired two-tailed t-test. Oligo oligomycin, FCCP Carbonyl cyanide 4-(trifluoromethoxy) phenylhydrazone, R/A rotenone, and antimycin A. **k** Immunoblots and protein quantification of glycolytic enzymes in P and Q MEFs. Representative immunoblots of four independent experiments are shown. HK hexokinase, PFK1 phosphofructokinase 1, ALDOA aldolase A, PKM pyruvate kinase M, LDHA lactate dehydrogenase A. Values are the mean ± SEM of 3–4 biologically independent experiments. *P* values by unpaired two-tailed student's *t*-test are indicated except for ****P* < 0.0001. **l** Immunoblots and protein quantification of mitochondrial pyruvate carriers and TCA cycle enzymes in P and Q MEFs. Vinculin was monitored as a loading control. Each immunoblot is representative of three independent experiments. MPC1-2 mitochondrial pyruvate carrier 1-2, IDH2 isocitrate dehydrogenase 2, DLST dihydrolipoamide S-succinyltransferase, SDHA succinate dehydrogenase, MDH2 malate dehydrogenase 2. Values are the mean ± SEM of 3–4 biologically independent experiments. *P* values by unpaired two-tailed student's *t*-test are indicated except for ****P* < 0.0001 and ns *P* = 0.1795. **m** Basal respiration and ATP production from OXPHOS measured by OCR of P and Q MEFs cultured in vehicle or 100 μM UK5099 for 24 h. Values are the mean ± SEM of 5–6 biologically independent experiments. *P* value by unpaired two-tailed student's *t*-test is indicated except for ****P* < 0.0001. **n** The cell death rate (%) determined with Hoechst-PI staining for P and Q MEFs cultured in vehicle or 100 μM UK5099 for 72 h. Values are the mean ± SEM of four biologically independent experiments; ns *P* = 0.2924, ****P* < 0.0001 using unpaired two-tailed *t*-test.

The expression of mitochondrial pyruvate carriers 1 and 2 (MPC1 and MPC2), the mitochondrial gatekeepers for pyruvate[26], IDH2, the rate-limiting enzyme of the TCA cycle, and some TCA cycle enzymes (DLST, SDHA, and fumarase) were upregulated in Q cells compared with P cells (Fig. 1l and Supplementary Fig. 3d). The marked increase in MPCs and TCA cycle enzymes together with the reduced expression of LDHA (Fig. 1k, l) could explain the dramatic increase in pyruvate translocation to mitochondria and the enhanced TCA cycle flux in Q cells. Upon MPC inhibition, both basal respiration and ATP production from the oxidative phosphorylation (OXPHOS) are significantly decreased in both P and Q cells (Fig. 1m). Overexpression of MPC1/2 in proliferating cells is sufficient to increase OXPHOS (Supplementary Fig. 3e, f). Thus, it appears that high MPC expression is the driver of increased OXPHOS. Interestingly, the survival of Q cells is dependent on MPC expression, as they are much more vulnerable to prolonged MPC inhibition than are P cells (Fig. 1n). Similar changes in the levels of metabolic enzymes and MPCs were also observed in Q primary MEFs, NIH3T3 cells, and human skin fibroblasts BJ cells (Supplementary Fig. 4a, b). Lastly, ECAR was reduced, and OCR was elevated in Q primary MEFs, NIH3T3, and BJ cells (Supplementary Fig. 4c, d). These results suggest that attenuated glycolysis and induced mitochondrial metabolism and respiration are common phenomena in quiescent fibroblasts. Interestingly, elevated MPC levels and OXPHOS and reduced glycolysis were also observed in Q Chinese hamster ovary (CHO) cells (Supplementary Fig. 4e–h), and the same was recently reported in Q neural stem cells[24].

**E-Cadherin and yes-associated protein (YAP) regulate glycolysis during the transition from a proliferative to a quiescent state**

Cell–cell interactions lead to cellular quiescence by contact inhibition[27,28]. Surprisingly, we found that E-cadherin (E-Cad) is induced when the cells reach confluence with a concomitant decrease in the expression of the rate-limiting glycolytic enzymes, HK2 and PKM2 (Fig. 2a). Interestingly, the induced expression of E-Cad to a different extent after quiescence was observed in independently isolated primary MEFs and immortalized MEFs by either dominant-negative p53 or SV40 large T cells (Supplementary Fig. 5a, b). Therefore, we hypothesized that cell–cell interactions through E-Cad could control intracellular glucose metabolism during the transition from a proliferative to a quiescent state. To determine whether E-Cad plays a role in metabolic programming, we deleted E-Cad and found that E-Cad deletion restored the ECAR in Q cells (Fig. 2b). In addition, the deletion of E-Cad restored the expression of HK2, PKM1, and PKM2 in Q cells, although p27$^{Kip1}$ levels were maintained at both the protein and mRNA levels (Fig. 2c and Supplementary Fig. 5c). $^{13}$C-MFA showed that glycolysis was rescued in Q E-Cad KO cells (Fig. 2d). This rescue is also reflected by a significantly higher abundance of (M + 2) pyruvate and lactate after [1,2-$^{13}$C$_2$]-glucose tracing in Q E-Cad KO cells (Fig. 2e, f). However, the flux of pyruvate transport to the mitochondria and the TCA cycle flux remained higher in Q E-Cad KO cells and was even higher than that in control Q cells (Supplementary Fig. 5d). Consistently, the OCR measurements showed that basal respiration was even higher in Q E-Cad KO cells than in control Q cells (Supplementary Fig. 5e). This result suggests that Q E-Cad KO cells are more energetic and produce ATP from both glycolysis and mitochondrial respiration. These results also suggest that E-Cad mediates the reduced glycolysis by inhibiting glycolytic enzyme expression in Q cells but not the increase in the TCA cycle and OXPHOS.

E-Cad was reported to drive contact inhibition and inhibition of cell proliferation via the Hippo-dependent signaling pathway and the inhibition of yes-associated protein (YAP) activity[29,30]. The Hippo pathway negatively regulates the nuclear localization and activity of the transcriptional coactivator yes-associated protein (YAP)[31]. P cells showed an increase in nuclear YAP localization and the expression of connective tissue growth factor (CTGF), a direct YAP target gene, whereas Q cells showed less total YAP intensity, probably due to degradation of cytosolic YAP along with significantly reduced CTGF expression (Supplementary Fig. 5f–h). YAP transcriptional activity, measured by 8XGTIIC (TEAD luciferase reporter)[32], was also significantly decreased in Q cells compared with P cells (Supplementary Fig. 5i). A similar YAP translocation pattern was observed in P versus Q primary MEFs (Supplementary Fig. 5j). Transcriptomic analysis consistently showed that the transcription of YAP target genes was markedly reduced in Q cells (Fig. 3a). Interestingly, in Q E-Cad KO cells, the nuclear localization of YAP was significantly enhanced (Fig. 3b, c), and concomitant YAP transcriptional activity as well as the mRNA levels of YAP target genes were restored (Fig. 3d and Supplementary Fig. 6a). Interestingly, Q E-Cad KO cells were arrested in the G$_1$ cell cycle phase (Supplementary Fig. 6b). Thus, although YAP activity was restored, Q E-Cad KO cells maintained their quiescent characteristics. Given that YAP is more transcriptionally active in P cells and can induce glycolytic enzyme expression[33,34], we hypothesized that cell–cell interactions through E-Cad control intracellular glucose metabolism by regulating YAP activity.

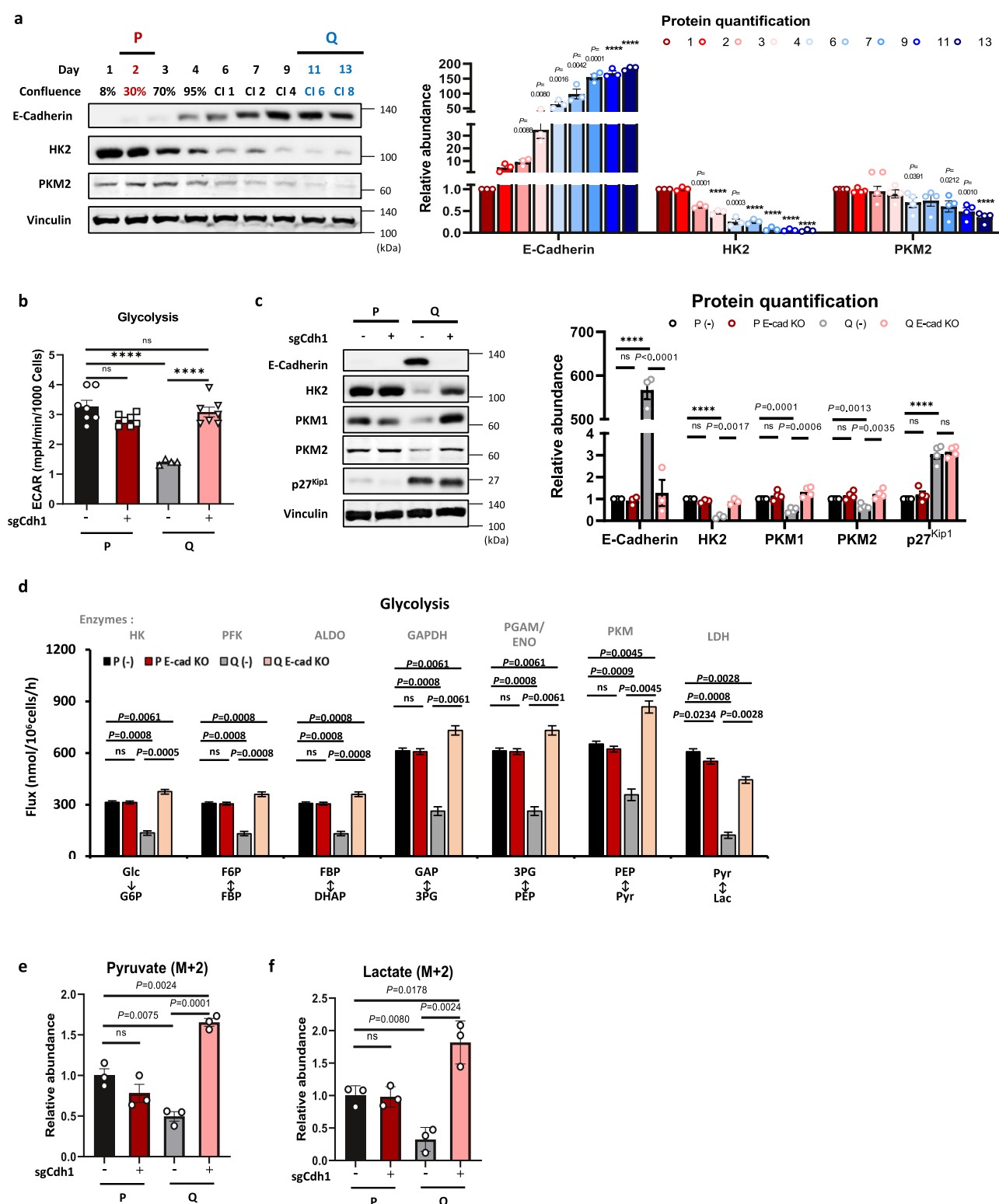

To determine whether YAP is responsible for metabolic reprogramming in Q cells, we knocked out TEAD, which is required for YAP and the PDZ-binding domain (TAZ) to interact with DNA. The 8XGTIIC (TEAD luciferase reporter) activity was significantly reduced in TEAD KO cells compared with sgTead1 (-) (wild-type) cells in the P state, and the activity was even lower in both conditions in the Q state (Fig. 3e). ECAR was reduced in P cells after TEAD KO (Fig. 3f) with a concomitant decrease in the expression of the rate-limiting enzymes of glycolysis, namely, HK2, PFK1, and PKM2, at the protein and mRNA levels (Fig. 3g

and Supplementary Fig. 6c). p27$^{Kip1}$ levels were not affected in both P and Q cells by TEAD KO (Fig. 3g).

To further investigate the effect of YAP, YAP knockout MEFs were generated, followed by the exogenous expression of YAP wild-type (WT) or a constitutively active YAP mutant (5SA)[35] (Supplementary Fig. 7a). As all possible inhibitory phosphorylation sites are mutated in YAP 5SA, it is constitutively localized to the nucleus in both P and Q cells (Supplementary Fig. 7b, c). YAP transcriptional activity was significantly reduced in Q WT cells compared with P WT cells, but its

**Fig. 2 | E-Cadherin and YAP regulate glycolysis during the transition from a proliferative to a quiescent state. a** Immunoblots and protein quantification of E-cadherin, hexokinase 2 (HK2), and pyruvate kinase M2 (PKM2) in MEFs from Day 1 to Day 13. Each immunoblot is representative of three independent experiments. Values are the mean ± SEM of 3–4 biologically independent experiments. *P* values by unpaired two-tailed student's *t*-test are indicated except for ****$P < 0.0001$. **b** Glycolytic function monitored by the extracellular acidification rate (ECAR) in control or E-cadherin KO MEFs in the P and Q states. Means ± SEMs ($n = 4$–7) are shown. ns (P E-cad KO $P = 0.0593$, Q E-cad KO $P = 0.7064$), ****$P < 0.0001$ using unpaired two-tailed *t*-test. **c** Immunoblots and protein quantification of E-cadherin, HK2, PKM1-2, and p27kip1 in control or E-cadherin KO MEFs in the P and Q states. Each immunoblot is representative of 3–4 independent. Means ± SEMs of 3-4 experiments are shown. ns (E-cad $P = 0.5084$, HK2 $P = 0.0807$, PKM1 $P = 0.2089$,

PKM2 $P = 0.1437$, P p27Kip1 $P = 0.2581$, Q p27Kip1 $P = 0.6608$), ****$P < 0.0001$ using unpaired two-tailed *t*-test. **d** Comparison of key intracellular glycolytic fluxes (estimated flux ± SD) in control or E-cadherin KO MEFs in the P and Q states. All flux fits passed the chi-square goodness-of-fit test. *P* values by unpaired two-tailed student's *t*-test are indicated except for ns (Glc → G6P $P = 0.8439$, F6P ↔ FBP $P = 0.7353$, FBP ↔ DHAP $P = 0.7353$, GAP ↔ 3PG $P = 0.7151$, 3PG ↔ PEP $P = 0.7151$, PEP ↔ Pyr $P = 0.1662$). **e, f** Relative abundance of intracellular 2x13C (M + 2) glycolytic metabolites (pyruvate and lactate) in control or E-cadherin KO MEFs in the P and Q states after 25 mM [1,2-13C2]-glucose labeling for 24 h. The area under the curve (AUC) was normalized to the levels of protein and internal standard. Means ± SEMs ($n = 3$) are shown. *P* values by unpaired two-tailed student's *t*-test are indicated except for ns (Pyruvate $P = 0.1875$, Lactate $P = 0.1875$).

---

activity was even higher in both P 5SA cells and Q 5SA cells (Fig. 4a). The mRNA levels of YAP target genes were also restored in Q 5SA cells (Supplementary Fig. 7d). The measurement of ECAR showed that glycolysis was reduced in Q WT cells but markedly increased in Q 5SA cells (Fig. 4b, c). In addition, the expression of glycolytic enzymes was restored in Q 5SA cells at both the protein and mRNA levels (Fig. 4d and Supplementary Fig. 7e). We confirmed that Q 5SA cells were arrested in the G1 cell cycle phase with elevated p27Kip1 levels and BrDU-PI staining (Fig. 4d and Supplementary Fig. 7e, f). Q 5SA cells had enhanced glucose uptake and lactate excretion levels that were higher than the levels of P 5SA cells (Fig. 4e and Supplementary Data 3). 13C-MFA revealed that while the glycolytic flux was reduced in Q WT compared with P WT cells, it was markedly induced in Q 5SA compared with P 5SA cells (Fig. 4f, g and Supplementary Data 2).

Interestingly, the flux to the PPP was silenced in Q 5SA cells, but the malate-to-pyruvate flux was increased almost 3-fold, probably to compensate for the reduced production of NADPH by the PPP (Fig. 4f). TCA cycle fluxes, which were markedly induced in Q WT cells, were further enhanced in Q 5SA cells (Fig. 4f and Supplementary Fig. 7g). Consistently, the basal respiration measured by the OCR was much higher in Q 5SA cells than in P cells (Supplementary Fig. 7h). These results are similar to the results obtained after the knockout of E-Cad in Q cells (Supplementary Fig. 5d, e). MPC1 and MPC2, as well as TCA cycle enzyme (IDH2 and SDHA) levels, were maintained at high levels in the Q state in all E-Cad KO, TEAD KO, and YAP 5SA cells (Supplementary Fig. 7i–k). Taken together, these results strongly indicate that in P cells, YAP elevates glycolysis by maintaining glycolytic enzyme expression, but in Q cells, YAP activity is attenuated, at least in part, by E-Cad and thus inhibits glycolytic enzyme expression and reduces glycolysis. However, the flux of pyruvate to mitochondria, the TCA cycle flux, and mitochondrial respiration remained elevated as well as the reversed MDH1 flux in Q cells regardless of YAP activity.

## The NAD+/NADH ratio is elevated in quiescent cells compared with proliferating cells largely because of elevated complex I activity

We found that the whole-cell NAD+/NADH ratio and cytosolic NAD+/NADH ratio were higher and that mitochondrial NADH was lower in Q cells than in P cells (Fig. 5a–c). In addition to the high level of malate derived from mitochondria, the high cytosolic NAD+/NADH ratio in Q cells could also contribute to the reversal of the MDH1 flux (from malate to oxaloacetate), as the reversal reaction could be driven by the relatively high level of NAD+[36]. Immunoblotting analysis showed that the MDH1 protein level was elevated in Q cells (Fig. 5d), which is consistent with the significantly higher absolute flux rate of malate to oxaloacetate (Fig. 1e). Given that aerobic glycolysis reflects increased cell demand for NAD+[37], the higher cytosolic NAD+/NADH ratio in Q cells could be explained, at least in part, by the lower demand for NAD+ in Q cells compared to highly glycolytic P cells. Indeed, increasing NAD+ in the cytosol by stable expression of cytoLbNOX elevated glycolysis only in P cells (Fig. 5e, f). However, an increase in mitochondrial

NAD+ by mitoLbNOX expression did not affect glycolysis in either P or Q cells. Surprisingly, we found that glycerol 3-phosphate dehydrogenase 1 (GPD1), which converts DHAP to glycerol 3-phosphate (G3P), was markedly elevated in Q cells (Supplementary Fig. 8a). Since GPD1 converts DHAP to G3P in a reaction that generates NAD+, its high expression in Q cells could also contribute to the higher cytosolic NAD+/NADH ratio.

Given that Q cells have higher mitochondrial membrane potential ($\Delta\Psi_m$) and basal respiration than those of P cells (Fig. 1g–j), we hypothesized that complex I activity could also be higher in Q cells and may have contributed to the increase in both the mitochondrial and whole-cell NAD+/NADH ratios. To determine that, we pharmacologically inhibited complex I. As expected, basal respiration was significantly attenuated following complex I inhibition (Fig. 5g). Inhibition of complex I with either rotenone or metformin significantly decreased the whole-cell NAD+/NADH ratio (Fig. 5h, Supplementary Fig. 8b). Complex I inhibition also significantly increased mitochondrial NADH and markedly decreased the mitochondrial membrane potential ($\Delta\Psi_m$) (Supplementary Fig. 8c–e). Taken together, these results suggest that hyperactive complex I in Q cells contributes to the higher whole-cell NAD+/NADH ratio in Q cells. Interestingly, in contrast to the whole-cell NAD+/NADH ratio, the cytosolic NAD+/NADH ratio was slightly elevated in P cells and was only slightly reduced in Q cells after complex I inhibition (Fig. 5i). This response is probably to support the elevated glycolysis after the inhibition of complex I in both P and Q cells (Fig. 5j). Indeed, glucose consumption was significantly increased even in Q cells by complex I inhibition (Supplementary Fig. 8f). A [U-13C] glucose tracing experiment revealed that the conversion of pyruvate to lactate by lactate dehydrogenase (LDH) was significantly enhanced by complex I inhibition in Q cells (Fig. 5k, l). Instead, the fractional enrichment of citrate (M + 2) was significantly reduced (Fig. 5m). Given that LDH generates cytosolic NAD+ as a byproduct and pyruvate dehydrogenase (PDH) consumes mitochondrial NAD+ (Fig. 5k), these results suggest that when mitochondrial NAD+ demand is increased by complex I inhibition, pyruvate is shunted to lactate instead of being metabolized in mitochondria.

## ME1 flux is suppressed in quiescent cells, whereas malate dehydrogenase 1 (MDH1) flux is reversed and elevated and phosphoenolpyruvate carboxykinase 1 (PCK1) flux is elevated

Interestingly, the ME1-mediated flux was diminished in Q cells (0 nmol/10^6 cells/h; from malate (Mal.c) to pyruvate (Pyr.c)), while it was active in P cells (31.6 nmol/10^6 cells/h) (Fig. 1e). The reduction in ME1-mediated flux cannot be explained by the reduced expression of ME1, as its expression level was similar in P and Q cells (Fig. 6a). The discrepancy between enzyme expression levels and metabolic flux by ME1 led us to assess whether there is any posttranslational modification regulating ME1 activity. As was previously reported, ACAT1-mediated K337 acetylation leads to ME1 dimerization and activation, and SIRT6-mediated K337 deacetylation leads to ME1 monomerization and deactivation (Fig. 6b)[38]. Therefore, to assess the multimerization of

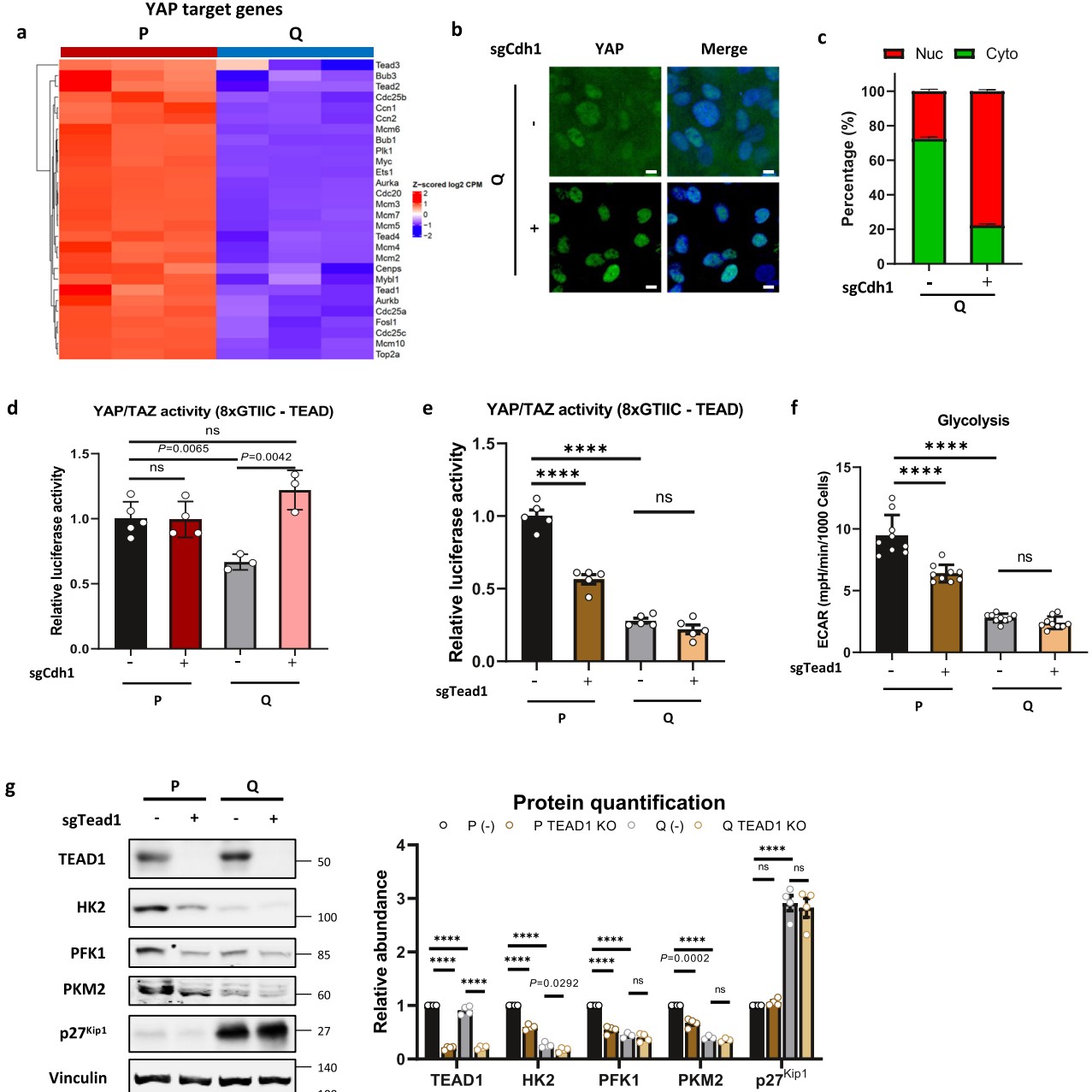

**Fig. 3 | TEAD deletion attenuates YAP transcriptional activity and glycolysis in P cells. a** Heatmap showing the expression of YAP target genes in P and Q MEFs as determined by RNA sequencing. Data represent values from three independent experiments, and Z-scored log2-fold change values are color-coded as indicated. **b**, **c** Localization analysis of YAP (green) in control or E-cadherin KO MEFs in the Q state by immunofluorescence. Hoechst 33342 (blue) was used to show the nuclei, and the scale bar size represents 10 μm. Each experiment included observation of at least 10 randomly selected fields (400× magnification). The quantified percentages (%) of nuclear (red) and cytosolic (green) YAP are indicated, and the data represent the mean ± SEM. The statistical significance of the differences was determined by two-way ANOVA; $P < 0.0001$. **d** Luciferase assays with the 8 × GTIIC-Lux reporter in control or E-cadherin KO MEFs in the P and Q states. Data are normalized to sgCdh1 (-) P cells (wild-type) and are presented as the mean ± SEM ($n = 3–5$). $P$ values by unpaired two-tailed student's t-test are indicated except for ns (P E-cad KO $P = 0.9571$, Q E-cad KO $P = 0.0713$). **e** Luciferase assays with the 8 × GTIIC-Lux

reporter in control or TEAD1 KO MEFs in the P and Q states. Data are normalized to sgTead1 (-) P cells (wild-type) and are presented as the mean ± SEM of $n = 5$ biologically independent samples. **** $P < 0.0001$, ns $P = 0.1242$ using unpaired two-tailed t-test. **f** Glycolytic function monitored by extracellular acidification rate (ECAR) in control or TEAD1 KO MEFs in the P and Q states. Data are presented as the mean ± SEM of $n \geq 9$ biologically independent samples. **** $P < 0.0001$, ns $P = 0.0744$ using unpaired two-tailed t-test. **g** Immunoblots and protein quantification of TEAD1, glycolytic enzymes, and p27$^{kip1}$ in control or TEAD1 KO MEFs in the P and Q states. Each immunoblot is a representative of four biologically independent experiments. Values are the mean ± SEM of 3–4 biologically independent experiments. $P$ values by unpaired two-tailed student's t-test are indicated except for **** $P < 0.0001$ and ns (PFK1 $P = 0.4610$, PKM2 $P = 0.1934$, P p27$^{kip1}$ $P = 0.1074$, Q p27$^{kip1}$ $P = 0.7180$). HK hexokinase, PFK1 phosphofructokinase 1, PKM2 pyruvate kinase M2. Values are the mean ± SEM of 3–4 biologically independent experiments.

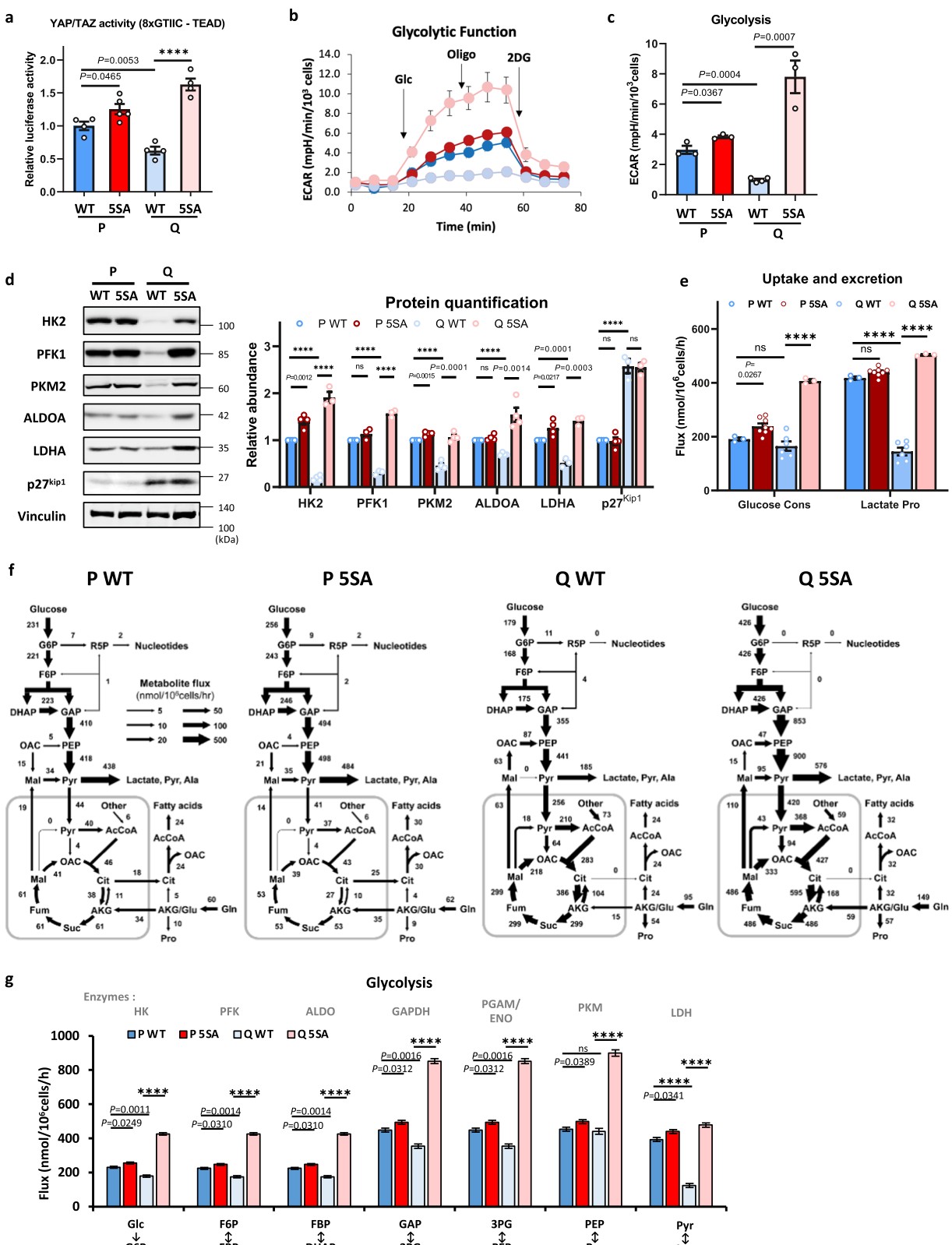

ME1, we treated both P and Q cells with bis(sulfosuccinimidyl)suberate (BS³), a water-soluble compound that has previously been used in the cross-linking of proteins[39]. Interestingly, treatment with BS³ resulted in the cross-linking of dimers in P cells but not in Q cells (Fig. 6c).

We recapitulated these results by expressing Flag-tagged and Myc-tagged ME1 followed by coimmunoprecipitation (Co-IP) with anti-Flag antibodies, which showed impaired dimerization with Myc-tagged ME1 in Q cells (Fig. 6d). The reduced dimerization of ME1 in Q cells is associated with increased binding of ME1 to SIRT6 and decreased binding to ACAT1 (Fig. 6d). In addition, acetylation of ME1, likely mediated by ACAT1, was higher in P cells than in Q cells (Fig. 6e). These data suggest that induced SIRT6-mediated deacetylation and reduced ACAT1-mediated acetylation result in decreased ME1 dimerization and activity in Q cells (Fig. 6f). The reaction catalyzed by ME1, which also

**Fig. 4 | Constitutively active YAP (YAP 5SA) expression rescues YAP transcriptional activity and glycolysis in Q cells. a** Luciferase assays with the 8×GTIIC-Lux reporter in WT and 5SA cells in the P and Q states. Data are normalized to WT P cells and are presented as the mean ± SEM of 4–5 biologically independent samples. *P* values by unpaired two-tailed student's *t*-test are indicated except for ****P* < 0.0001. **b**, **c** The extracellular acidification rate (ECAR) and individual glycolysis parameters in WT and 5SA cells in the P and Q states. Data are presented as the mean ± SEM of 3–4 biologically independent samples. *P* values by unpaired two-tailed student's *t*-test are indicated. Glc glucose, Oligo oligomycin, 2DG 2-deoxyglucose. **d** Immunoblots and protein quantification of glycolytic enzymes and p27^kip1 in WT and 5SA cells in the P and Q states. Vinculin was monitored as a loading control. Representative immunoblots of four independent experiments are shown. Values are the mean ± SEM of 3-4 biologically independent experiments. *P* values by unpaired two-tailed student's *t*-test are indicated except for ****P* < 0.0001

and ns (PFK1 *P* = 0.0667, ALDOA *P* = 0.1451, P p27^kip1 *P* = 0.7940, Q p27^kip1 *P* = 0.9048). HK hexokinase, PFK1 phosphofructokinase 1, PKM pyruvate kinase M, ALDOA aldolase A, LDHA lactate dehydrogenase A. **e** Comparison of the quantified key extracellular metabolic fluxes (estimated flux ± SD) in WT and 5SA cells in the P and Q states. The flux rates of glucose uptake and lactate excretion are plotted. *P* values by unpaired two-tailed student's *t*-test are indicated in the figure except for ****P* < 0.0001 and ns *P* values (Glucose Cons. Q WT *P* = 0.3440, Lactate Pro P 5SA *P* = 0.0557) **f** Metabolic flux maps of WT and 5SA cells in the P and Q states. The flux maps were determined using ¹³C-MFA of multiple datasets at isotopic steady state as described in Methods. **g** Comparison of key intracellular glycolytic fluxes (estimated flux ± SD) in WT and 5SA cells in the P and Q states. All flux fits passed the chi-square goodness-of-fit test. *P* values by unpaired two-tailed student's *t*-test are indicated except for ns *P* = 0.5742.

generates NADPH, might be suppressed in Q cells because they may not require high NADPH levels, and NADPH generated by the PPP in these cells might be sufficient. Indeed, both cytosolic NADPH and NADP⁺ levels measured by the iNap1 sensor[40] were significantly lower in Q cells than in P cells, and the cytosolic NADP⁺/NADPH ratio was higher in Q cells (Supplementary Fig. 9a–c). Interestingly, although the glycolytic fluxes were reduced in Q cells, the PPP flux was not reduced. Since the major source of NADPH in Q cells is the PPP, it could explain previous results showing that Q cells are sensitive to the inhibition of the PPP[12].

The ¹³C-MFA also revealed that P cells and Q cells have an opposite direction of net malate dehydrogenase 1 (MDH1)-mediated flux, which is typically considered reversible[36]; while P cells have a downward flux from oxaloacetate to malate (OAC.c → Mal.c, 24.4 nmol/10⁶ cells/h), Q cells have a significantly upregulated upward flux from malate to oxaloacetate (Mal.c → OAC.c, 86.6 nmol/10⁶ cells/h) (Figs. 1e and 6g, h and Supplementary Data 2). As indicated above, this could be explained by the high levels of malate derived from mitochondria (Mal.m → Mal.c), by the relatively high level of cytosolic NAD⁺, and by the upregulated MDH1 expression in Q cells (Figs. 1e, 5b, 6a and Supplementary Fig. 9d). In addition, the relatively low levels of cytosolic OAC (OAC.c) derived from mitochondria in Q cells compared to P cells (Fig. 1e and Supplementary Fig. 9e) could also contribute to the upward flux from malate to oxaloacetate (Mal.c → OAC.c) in Q cells. Indeed, the major source of OAC.c in P cells was cytosolic citrate (Cit.c), the majority of which seemed to be derived from mitochondria (Cit.m) (Fig. 1e and Supplementary Fig. 9f). However, in Q cells, OAC.c was mainly produced from cytosolic malate (Mal.c) by enhanced MDH1 flux (Fig. 6g–h); in contrast, OAC.c derived from Cit.c was suppressed probably caused by the diminished flux from Cit.m to Cit.c (Fig. 1e and Supplementary Fig. 9f).

The phosphoenolpyruvate carboxykinase 1 (PCK1)-mediated flux was significantly increased in Q cells (104 nmol/10⁶ cells/h; from OAC.c to phosphoenolpyruvate (PEP.c)) compared with P cells (7.4 nmol/10⁶ cells/h) (Figs. 1e and 6g, h). This is mediated by the high level of OAC.c generated by MDH1 and by the induced expression of PCK1 (Fig. 6a). Consequently, PEP.c is produced at high levels, independent of glycolysis, to generate sufficient pyruvate that is shuttled to mitochondria. Collectively, these results suggest that, through the upregulation of MDH1 (reversed) and PCK1 fluxes, Q cells recycle malate derived from the TCA cycle (Mal.m) and regenerate pyruvate in cytosol (Pyr.c). This process additionally facilitates the TCA cycle and mitochondrial respiration.

**Quiescent cells suppress the malate-aspartate shuttle (MAS), whereas proliferating cells are dependent on it for cell proliferation**

To maintain a high cytosolic NAD⁺/NADH ratio necessary for sustained active glycolysis, highly glycolytic cells employ several strategies. Firstly, these cells activate lactate dehydrogenase (LDH), utilizing

NADH to catalyze the reduction of pyruvate to lactate, thereby replenishing the NAD⁺ pool[41]. Alternatively, NADH electrons in the cytosol can be transferred to the mitochondria via the malate-aspartate shuttle (MAS) and the glycerol 3-phosphate shuttle (G3PS). These shuttles rely on specific cytosolic enzymes, namely malate dehydrogenase 1 (MDH1) and glycerol 3-phosphate dehydrogenase 1 (GPD1), respectively[42,43].

Since glycolytic activity was decreased, the reaction catalyzed by MDH1 is reversed, and the major source of cytosolic malate (Mal.c) is exported from mitochondria (Mal.m) in Q cells (Figs. 1c–e and 6g and Supplementary Fig. 9d), we expected that the MAS may not play a role in regulating the NAD⁺/NADH ratio in Q cells. Indeed, Q cells show reduced expression of SLC25A11 (also known as oxoglutarate carrier, OGC) and SLC25A13 (aspartate-glutamate carrier, AGC), which are transporters involved in the MAS (Fig. 7a, b). Consistently, when we treated cells with the MAS inhibitor aminooxyacetic acid (AOA)[44] or deleted SLC25A11, we found that it significantly reduced the cytosolic NAD⁺/NADH ratio and mitochondrial NADH only in P cells (Fig. 7c–e and Supplementary Fig. 10a). These results show that P cells are more dependent on the MAS than are Q cells for NAD⁺/NADH homeostasis.

Interestingly, MAS inhibition significantly slowed the proliferation of P cells (Fig. 7f). As MAS inhibition reduced glycolysis in both P and Q cells (Supplementary Fig. 10b), we hypothesized that the delayed cell growth by MAS inhibition was due to the reduced cytosolic NAD⁺/NADH ratio, which can slow glycolysis. Therefore, we supplemented the cells with cell-permeable pyruvate to upregulate cytosolic NAD⁺ by LDH-mediated flux (from pyruvate to lactate). Indeed, pyruvate supplementation rescued the proliferation, cytosolic NAD⁺/NADH ratio, and glycolysis (Fig. 7f–i). These data suggest that satisfying cytosolic NAD⁺ demand is critical for optimal levels of glycolysis and proper cell growth in P cells.

**Transcriptomic reprogramming in the transition from a proliferative to a quiescent state**

The transcriptomic analysis showed that Q cells were transcriptionally active, and surprisingly, large number of genes were elevated; among 13,022 significantly differentially expressed genes, 7131 genes were upregulated, and 5891 genes were downregulated in Q cells (Supplementary Data 1). The heatmap, using stricter filtering criteria (q 2), shows that out of 3319 genes, 2478 were upregulated and 841 were downregulated in Q cells (Supplementary Fig. 1a). From a total of 1251 Reactome pathway database gene sets considered, 28 were upregulated and 240 were downregulated in Q/P (FDR ≤ 0.05). The top 30 down- and 28 upregulated gene sets by significance are shown (Supplementary Fig. 1b). Upregulated gene sets were related to extracellular matrix (ECM) formation and interactions (Supplementary Fig. 1b). Specifically, gene sets associated with collagen biosynthesis and modifying enzymes, collagen trimerization, ECM proteoglycans, keratin sulfate biosynthesis and molecules associated with elastic fibers were significantly upregulated in Q cells. On the other hand,

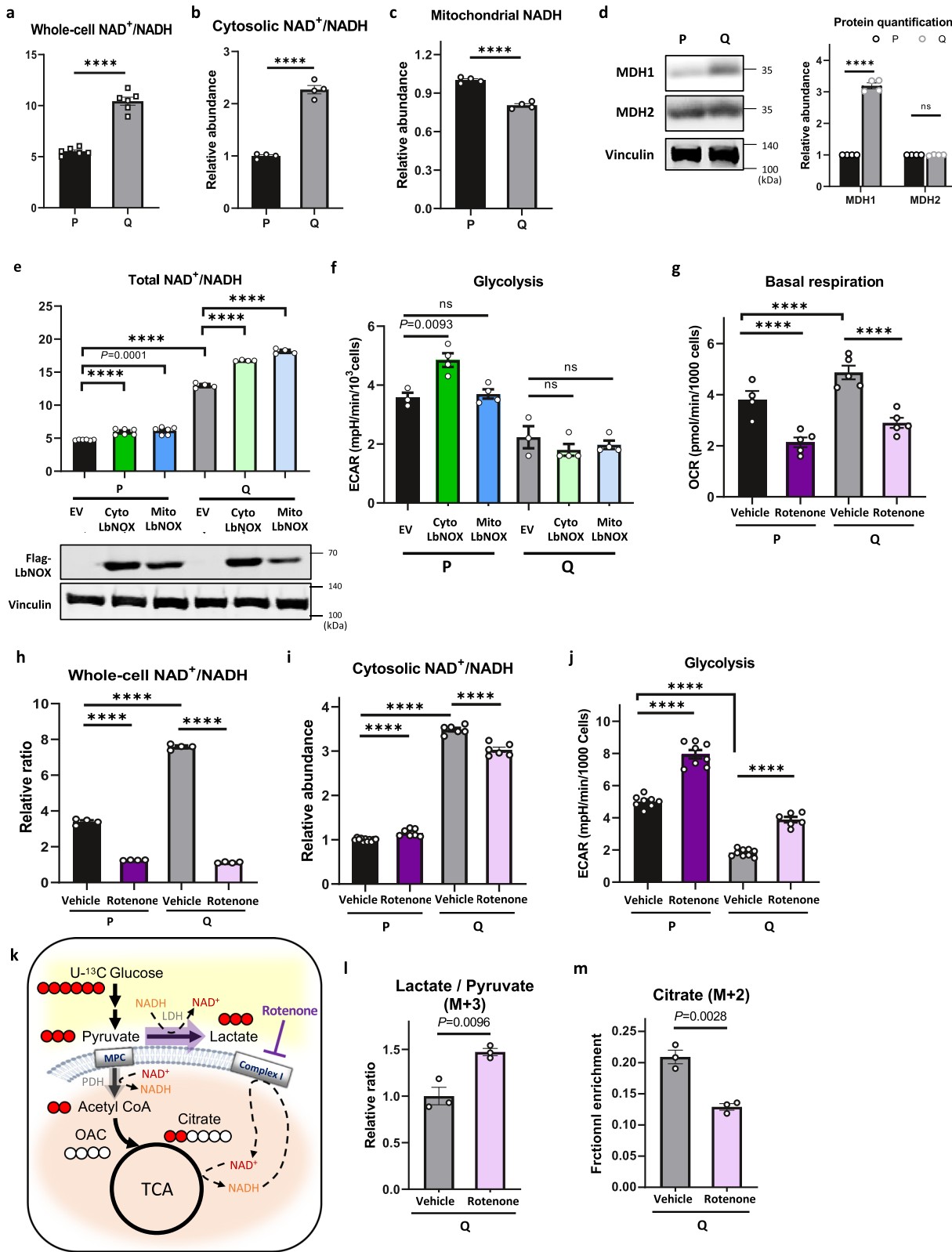

downregulated gene sets included those associated with cell cycle checkpoints, M phase, DNA replication and repair and S phase (Supplementary Fig. 1b). Among the genes that were suppressed at the mRNA level in Q cells are YAP target genes (Fig. 3a), G₂M checkpoint-related genes (Supplementary Fig. 1b–d), and glycolysis-related genes (Supplementary Fig. 2f, g). Interestingly, some genes are regulated by

posttranscriptional mechanisms. For example, GPD1, which is markedly upregulated at the protein level in Q cells (Supplementary Fig. 8a), did not appear to be upregulated at the mRNA level (Supplementary Data 1). SLC25A11, which is downregulated at the protein level in Q cells (Fig. 7a), did not appear to be downregulated at the mRNA level (Supplementary Data 1).

**Fig. 5 | The NAD⁺/NADH ratio is elevated in Q cells compared with P cells because of elevated complex I activity. a** Whole-cell NAD⁺/NADH ratio of P and Q MEFs. Data are presented as the mean ± SEM of 6 biologically independent samples. ****$P < 0.0001$ using unpaired two-tailed $t$-test. **b** Cytosolic NAD⁺/NADH ratio of P and Q MEFs measured by the pMOS023: Peredox NADH/NAD⁺ sensor (cytosolic) using flow cytometry. The mean MFI was calculated based on 4 biologically independent experiments. Data are presented as the mean ± SEM. ****$P < 0.0001$ using unpaired two-tailed $t$-test. **c** The relative abundance of mitochondrial NADH in P and Q MEFs measured by the pC1-mitoRexYFP sensor using flow cytometry. The mean MFI was calculated based on 4 biologically independent experiments. Data are presented as the mean ± SEM. **** $P < 0.0001$ using unpaired two-tailed $t$-test. **d** Immunoblots and protein quantification of MDH1 and MDH2 in P and Q MEFs. Values are the mean ± SEM of 4 biologically independent experiments. ****$P < 0.0001$, ns $P = 0.3892$ using unpaired two-tailed $t$-test. **e** The whole-cell NAD⁺/NADH ratio in either empty vector (EV), cytosolic (Cyto) or mitochondrial (Mito) LbNOX-expressing P or Q MEFs. Representative immunoblots confirm Flag-tagged LbNOX expression. Data are presented as the mean ± SEM of 4-6 independent experiments. $P$ value by unpaired two-tailed student's $t$-test is indicated except for ****$P < 0.0001$. **f** Glycolysis measured by the extracellular acidification rate (ECAR) in EV-, CytoLbNOX-, and MitoLbNOX-expressing cells in P and Q MEFs. Data are presented as the mean ± SEM of 3-4 biologically independent samples. $P$ value by unpaired two-tailed student's $t$-test is indicated except for ns (P MitoLbNOX $P = 0.5999$, Q CytoLbNOX $P = 0.3209$, Q MitoLbNOX $P = 0.5020$). **g** Basal respiration measured by oxygen consumption rate (OCR) of P and Q MEFs cultured in vehicle or 0.2 μM rotenone for 24 h. Values are the mean ± SEM of 4–5 independent experiments. ****$p < 0.0001$ using unpaired two-tailed $t$-test. **h** The whole-cell NAD⁺/NADH ratio in P or Q MEFs cultured in vehicle or 0.2 μM rotenone for 24 h. Data are presented as the mean ± SEM of 4 biologically independent samples. ****$P < 0.0001$ using unpaired two-tailed $t$-test. **i** The cytosolic NAD⁺/NADH ratio in P or Q MEFs cultured in vehicle or 0.2 μM rotenone for 24 h. Data are presented as the mean ± SEM of 6 biologically independent samples. ****$P < 0.0001$ using unpaired two-tailed $t$-test. **j** Glycolysis measured by the extracellular acidification rate (ECAR) of P and Q MEFs cultured in vehicle or 0.2 μM rotenone for 24 h. Data are presented as the mean ± SEM of 6–8 biologically independent samples. ****$P < 0.0001$ using unpaired two-tailed $t$-test. **k** Simplified schematic of the ¹³C labeling patterns of metabolites in glycolysis and the TCA cycle with [U-¹³C] glucose tracing (Created with Biorender.com released under a Creative Commons Attribution-NonCommercial-NoDerivs 4.0 International license). Red fills indicate ¹³C-labeled carbons. The mitochondrial pyruvate carrier (MPC) and Complex I are depicted on the mitochondrial membrane. **l** Relative ratio of (M + 3) lactate and pyruvate after 25 mM [U-¹³C] glucose labeling in Q MEFs cultured with vehicle or 0.2 μM rotenone for 24 h. Means ± SEMs ($n = 3$) are shown. $P$ value by unpaired two-tailed student's $t$-test is indicated. **m** Relative abundance of intracellular (M + 2) citrate after 25 mM [U-¹³C] glucose labeling in Q MEFs cultured with vehicle or 0.2 μM rotenone for 24 h. Means ± SEMs ($n = 3$) are shown. $P$ value by unpaired two-tailed student's $t$-test is indicated.

## Active mitochondrial metabolism supports extracellular matrix (ECM) biosynthesis in quiescent cells

To understand whether the metabolic changes we observed are required for quiescence, we revisited the transcriptomic analysis encompassing the entire gene expression spectrum. As previously highlighted, GSEA identified ECM formation and interactions as significantly enriched pathways in Q cells (Supplementary Fig. 1b). *De-novo* mRNA translation and secretion of ECM proteins requires ATP, which is generated by the marked increase of OXPHOS in quiescent cells. In addition to ATP production, the TCA cycle generates essential metabolites for macromolecule synthesis[45], with amino acid metabolism particularly reliant on mitochondrial enzymes[46]. The expression of genes associated with the collagen family, a key ECM component[47], are markedly upregulated in Q cells compared with P cells (Fig. 8a–c and Supplementary Data 1). We confirm the induction of collagen genes at the protein levels as well (Fig. 8d). Collagen family proteins are highly enriched in proline, which is exclusively generated by mitochondria from glutamate[14]. It was previously shown that proline is required for collagen synthesis in fibroblasts[48–50]. Indeed, we found that Q cells exhibit elevated flux of proline synthesis (Fig. 8e, and Supplementary Data 2 and 3). Finally, pyrroline-5-carboxylate synthase (P5CS) and pyrroline-5-carboxylate reductase 1 (PYCR1), the key mitochondrial enzymes responsible for proline biosynthesis, are significantly induced in Q cells, with PYCR1 being more robustly induced (Fig. 8f, g). The inhibition of MPCs or complex I which decrease OXPHOS and ATP production (Figs. 1m and 5g) reduced the expression of collagen proteins (Fig. 8h, i), which is consistent with previous results showing that mitochondrial activity is required for the generation of proline to synthesize collagens[50]. These findings collectively suggest that the augmented TCA cycle and OXPHOS in Q cells serve not only to elevate ATP production but also contribute to enhanced amino acid synthesis, thereby fortifying the production of ECM. Of note, as the main roles of fibroblasts is to produce and maintain turnover of the ECM, this underscores the necessity for sustained robust mitochondrial metabolism to effectively support these essential cellular functions.

## Discussion

The metabolic and transcriptomic reprogramming that occurs during the transition of cells from a proliferative to a quiescent state has not been thoroughly studied previously. Here, our ¹³C-metabolic flux analysis (¹³C-MFA) and transcriptomic analysis in MEFs provided evidence that Q cells are metabolically more active at the mitochondrial level than P cells and that they are also transcriptionally active. Our metabolic flux analysis showed that glycolytic fluxes are decreased but that the PPP is maintained in Q cells. We showed that the decrease in the glycolytic flux in Q cells is culminated in a decrease in lactate excretion. The decrease in lactate excretion is much more pronounced than the decrease in glycolytic fluxes because pyruvate is shunted to the TCA cycle instead of to lactate production. The decrease in glycolysis is associated with a decrease in glycolytic enzyme expression. These decreases in glycolytic enzyme expression and the glycolytic flux rate are largely dependent on YAP activity, which is diminished in Q cells. We showed that YAP activity is reduced in Q cells because of increased E-Cad expression. However, ECM proteins such as collagen, which are also induced in Q cells, could also in principle inhibit YAP activity[51]. We also showed that the expression of constitutively active YAP (YAP 5SA) dramatically increased glycolysis in Q cells.

Surprisingly, we found that the flux of the TCA cycle and OXPHOS were markedly increased in Q cells in a YAP-independent manner. In Q cells, pyruvate is shunted to the TCA cycle instead of to lactate production, which is associated with reduced LDH expression and a marked increase in the expression of the mitochondrial pyruvate transporters MPC1 and MPC2. Indeed, Q cells are specifically vulnerable to prolonged inhibition of MPC1 and MPC2, as this inhibition-induced cell death of Q cells but not of P cells. Thus, Q cells are metabolically active while not dividing but generate ATP primarily through the increase in OXPHOS. Mechanistically, pyruvate is shunted to the mitochondria instead of to lactate in Q cells because of induced MPCs expression and reduced demand for cytosolic NAD⁺. Interestingly, it was recently reported that in quiescent neural stem/progenitor cells, MPCs expression is induced as well as mitochondrial metabolism[24]. Thus, the increases in MPC levels and mitochondrial metabolism are not limited to quiescent MEFs and might be a common feature of quiescent cells. It should be noted that we analyzed Q cells after contact inhibition in the presence of growth factors. However, quiescence can also be acquired when cells are deprived of growth factors. When we deprived MEFs of growth factors, we found that both glycolysis, as measured by ECAR, and OXPHOS, as measured by OCR were decreased (Supplementary Fig. 11). This suggests that growth factors might be required for contact-inhibited Q cells to elevate the TCA cycle and OXPHOS.

The oxidative PPP appears to be the primary source of cytosolic NADPH in Q cells because ME1, which generates cytosolic NADPH by converting malate to pyruvate, is not active in these cells. This is

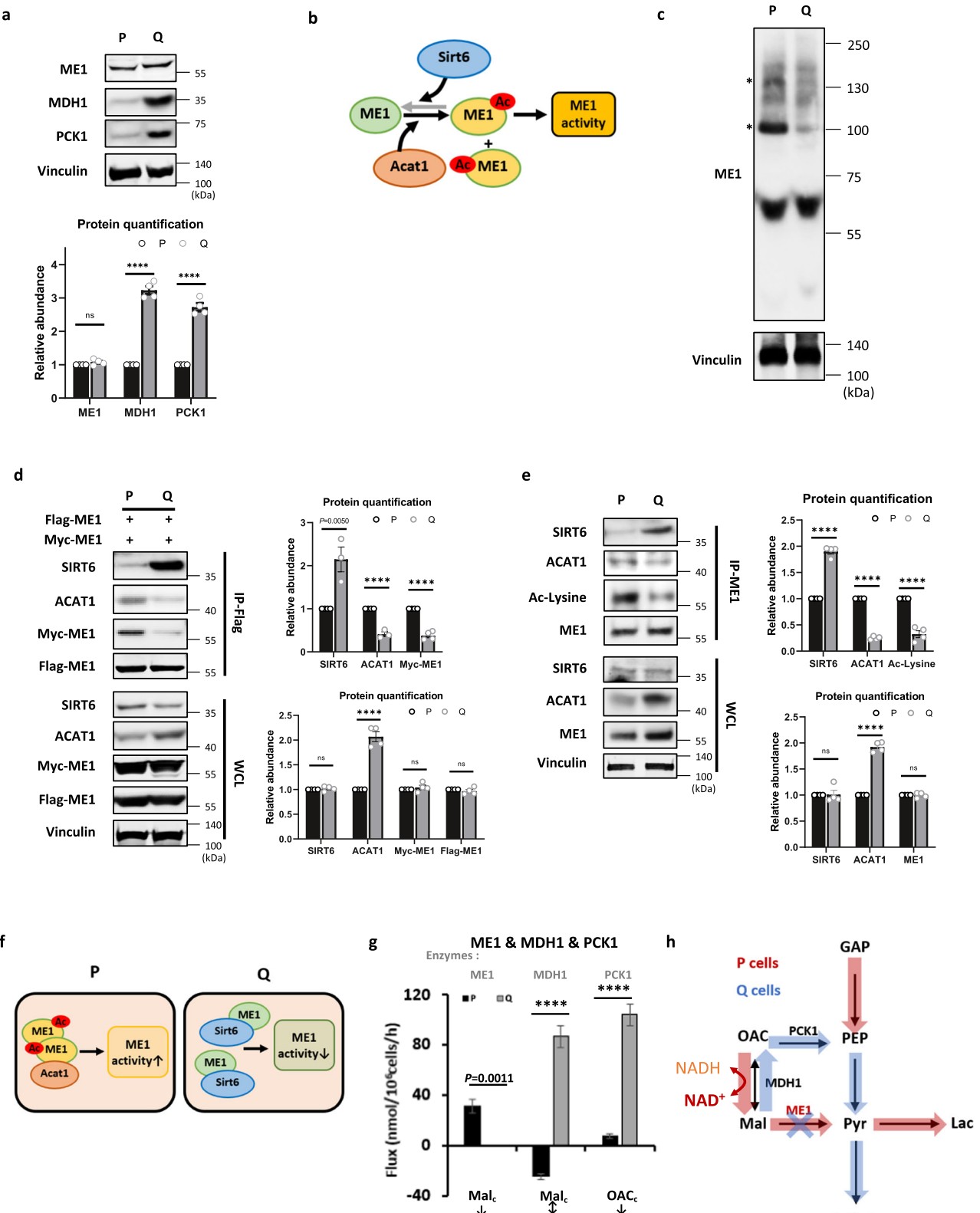

consistent with previous results showing that Q cells are dependent on the PPP for their survival[12]. Mechanistically, this phenomenon could be mediated by deacetylation of ME1 by SIRT6 and monomerization. Since NAD[+] is an essential cofactor for sirtuins[52], a relatively high cytosolic NAD[+]/NADH ratio might have contributed to the high SIRT6 activity in Q cells. Another intriguing observation is that the conversion of malate to oxaloacetate catalyzed by MDH1 is markedly increased in Q cells, suggesting that the MAS is not operating or is working in reverse in Q cells.

Consistently, we found that the expression levels of the mitochondrial transporter SLC25A11, which transports malate from the cytosol to the mitochondria, and SLC25A13, which transports aspartate from the mitochondria to the cytosol, were decreased in Q cells.

Intriguingly, [13]C-metabolic flux analysis revealed that the major source of cytosolic oxaloacetate (OAC.c) is different between P and Q cells. In P cells, OAC.c is mainly produced from cytosolic citrate (Cit.c). However, in Q cells, OAC.c is mainly derived from cytosolic malate

**Fig. 6 | ME1 dimerization and ME1-mediated flux are suppressed, whereas the fluxes through MDH1 and PCK1 are elevated in Q cells. a** Immunoblots and protein quantification of ME1, MDH1, and PCK1 in P and Q MEFs. Vinculin was used as a loading control. Values are the mean ± SEM of 4 biologically independent experiments. ns $P = 0.1412$, ****$P < 0.0001$ using unpaired two-tailed $t$-test. ME1 malic enzyme 1, MDH1 malate dehydrogenase 1, PCK1 phosphoenolpyruvate carboxykinase 1. **b** Schematic of ME1 acetylation by ACAT1 and deacetylation by SIRT6. **c** BS³ cross-linking assay in P and Q MEFs. After the addition of BS³, shifted bands that indicated protein–protein cross-linking could be seen; the shifted bands are marked with an asterisk. Representative data from six independent experiments are presented. **d** Coimmunoprecipitation (Co-IP) assay shows that ME1 dimerization is enhanced in P cells. Both Flag-tagged ME1 and Myc-tagged ME1 were expressed in MEFs, and the extracts were subjected to Co-IP with an anti-Flag tag antibody. Both immunoprecipitates with Flag (IP-Flag) and whole-cell lysates (WCL) were analyzed by Western blotting and protein quantification with means ± SEMs ($n = 3$–4) are shown. $P$ value by unpaired two-tailed student's $t$-test is indicated in the figure except for ****$P < 0.0001$ and ns (SIRT $P = 0.4538$, Myc-ME1 $P = 0. 3851$, Flag-ME1 $P = 0.3780$). **e** Co-IP assay shows that ME1 acetylation is enhanced in P cells. Extracts were subjected to Co-IP with the anti-ME1 antibody. Both immunoprecipitates with endogenous ME1 (IP-ME1) and whole-cell lysates (WCL) were analyzed by Western blotting and protein quantification with means ± SEMs ($n = 3$–4) are shown. ****$P < 0.0001$ and ns (SIRT $P = 0.9095$, ME1 $P = 0.8684$) using unpaired two-tailed $t$-test. **f** Schematic of ME1 acetylation and dimerization in P and Q MEFs. **g** Comparison of the quantified ME1-, MDH1-, and PCK1-mediated fluxes (estimated flux ± SD) in P and Q MEFs. P cells are shown in black bars, and Q cells are shown in gray bars. All flux fits passed the chi-square goodness-of-fit test. $P$ values by unpaired two-tailed student's $t$-test are indicated except for ****$P < 0.0001$. **h** Schematic of ME1-, MDH1- and PCK1-mediated fluxes in P and Q MEFs.

(Mal.c) exported from mitochondria (Mal.m) and converted to OAC.c by MDH1. This notion is supported by the high cytosolic $NAD^+/NADH$ ratio in Q cells, as relatively high $NAD^+$ levels can drive MDH1 to convert Mal.c to OAC.c. In P cells, the import of malate into mitochondria from the cytosol was active (Mal.c → Mal.m), which is mediated by the active MAS to support the high cytosolic $NAD^+$ demand. However, as Q cells have reduced cytosolic $NAD^+$ demand and MAS activity, the flux rate of malate exported from the mitochondria to the cytosol (Mal.m → Mal.c) is higher than the flux rate of malate import into mitochondria (Mal.c → Mal.m). Much more malate production in mitochondria from the active TCA cycle may also have contributed to this flux in Q cells.

Importantly, the deletion of E-Cad or expression of 5SA mutant of YAP did not inhibit quiescence. Both E-Cad deletion and the expression of YAP 5SA elevate glycolytic fluxes. However, they also further elevate the TCA cycle and OXPHOS. Therefore, we speculate that the metabolic hallmark of quiescent cells is a high ratio of OXPHOS to glycolysis and a reversed MDH1-mediated flux.

Transcriptomic reprogramming in Q cells is partially mediated by YAP. In agreement with previous results with human skin fibroblasts[12], we observed a dramatic induction of ECM-associated genes in Q cells. It is highly possible that many of the mRNA changes in Q cells are due to changes in mRNA stability, as was previously shown in human skin fibroblasts[53]. We also observed some discrepancy between the changes in protein expression levels and the changes in mRNA levels, suggesting that some of the changes in protein expression in Q cells occur at the translational or protein stability level. The inhibition of MPC or inhibition of complex I by rotenone, which inhibits OXPHOS reduced the expression of collagens. Therefore, we propose that high TCA cycle flux and OXPHOS in quiescent cells are required to maintain *de-novo* ECM protein synthesis by generating ATP and amino acids, especially proline, which is enriched in collagen family proteins. The secretion of ECM proteins is also a process that demands ATP. In summary, our findings underscore the metabolically and transcriptionally active nature of quiescent fibroblasts, despite their non-proliferative status, and illuminate the occurrence of specific metabolic reprogramming events. Physiologically, it remains to be determined if contact-inhibited fibroblasts in the body or quiescent cancer-associated fibroblasts undergo similar reprogramming process.

## Methods

List of plasmids, antibodies, reagents, cell lines, and primers used can be found in Supplementary Table 1.

### Cell culture

Mouse embryonic fibroblasts (MEFs) were isolated and immortalized as previously described[15,16] and were cultured in DMEM (Corning) containing 10% FBS (Gemini) and 1% penicillin/streptomycin (Corning). NIH 3T3 cells, BJ cells, and Chinese hamster ovary (CHO) cells were obtained from the American Type Culture Collection and were grown according to recommendations. Cells were routinely checked for mycoplasma contamination with an in-house PCR method. Cells were not used for more than 10 passages.

### Proliferating cells (P) and quiescent cells (Q)

The cells were analyzed while actively proliferating (2 days after plating) and when they were quiescent (contact-inhibited for 6 days, 11 days after plating) for MEFs, primary MEFs, NIH 3T3, BJ, and Chinese hamster ovary (CHO) cells. Cell culture medium was changed every other day for both proliferating and quiescent states. All assays were done 24 h after changing medium unless it is indicated.

### Cloning

Each sgRNA sequence was inserted into the lentiCRISPR v2 vector-Blast (Addgene plasmid #83480) or lentiCRISPR v2 vector-Hygro (Addgene plasmid # 98291) according to the recommended protocol on Addgene's webpage. sgRNAs were designed on CRISPOR.tefor.net or were taken from previously published works. For transfection of sgRNA plasmids in MEFs, $3 \times 10^5$ cells were plated in 6-well plates one day before transfection using 3 μg of each sgRNA plasmid. After 24 h of transfection, selection was carried out with blasticidin (10 μg/ml) or hygromycin (200 μg/ml). An untransduced negative control plate was used to determine when selection was performed. The cDNA for human YAP WT (pBABEpuro-Flag-YAP2, Addgene plasmid #27472) and the constitutively active mutant YAP 5SA (pCMV-flag YAP2 5SA, Addgene plasmid #27371) were cloned into the pBABE Hygro vector using Gibson Assembly according to the manufacturer's protocol (NEB).

### Lentivirus and retrovirus production, transduction, and antibiotic selection

For lentivirus production, each lentiviral plasmid was cotransfected with 9 μg of Virapower packing mix in 293FT cells on poly-L-lysine (Sigma)-coated 10-cm plates using Lipofectamine 2000 as described in a protocol on the Addgene website. Lentivirus infection was performed overnight in the presence of polybrene, and selection was carried out with blasticidin (10 μg/ml) or hygromycin (200 μg/ml). An untransduced negative control plate was used to determine when selection was performed.

For retrovirus production, Phoenix-AMPHO cells were transfected at 70–80% confluency on a 10-cm plate. Twelve micrograms of pBabe plasmid with 6 μg of VSVG was packaged in 60 μl of Lipofectamine 2000 in 3 ml of optiMEM with 10 ml of complete medium. 16 h after transfection, the medium was aspirated, and 10 ml of complete medium was added. 48 h later, the medium was collected, centrifuged, and filtered through a 0.45-μm filter (Millipore). Retrovirus infection was performed during 48 h of proliferation, and selection was carried out with hygromycin (200 μg/ml). An untransduced negative control plate was used to determine when selection was performed.

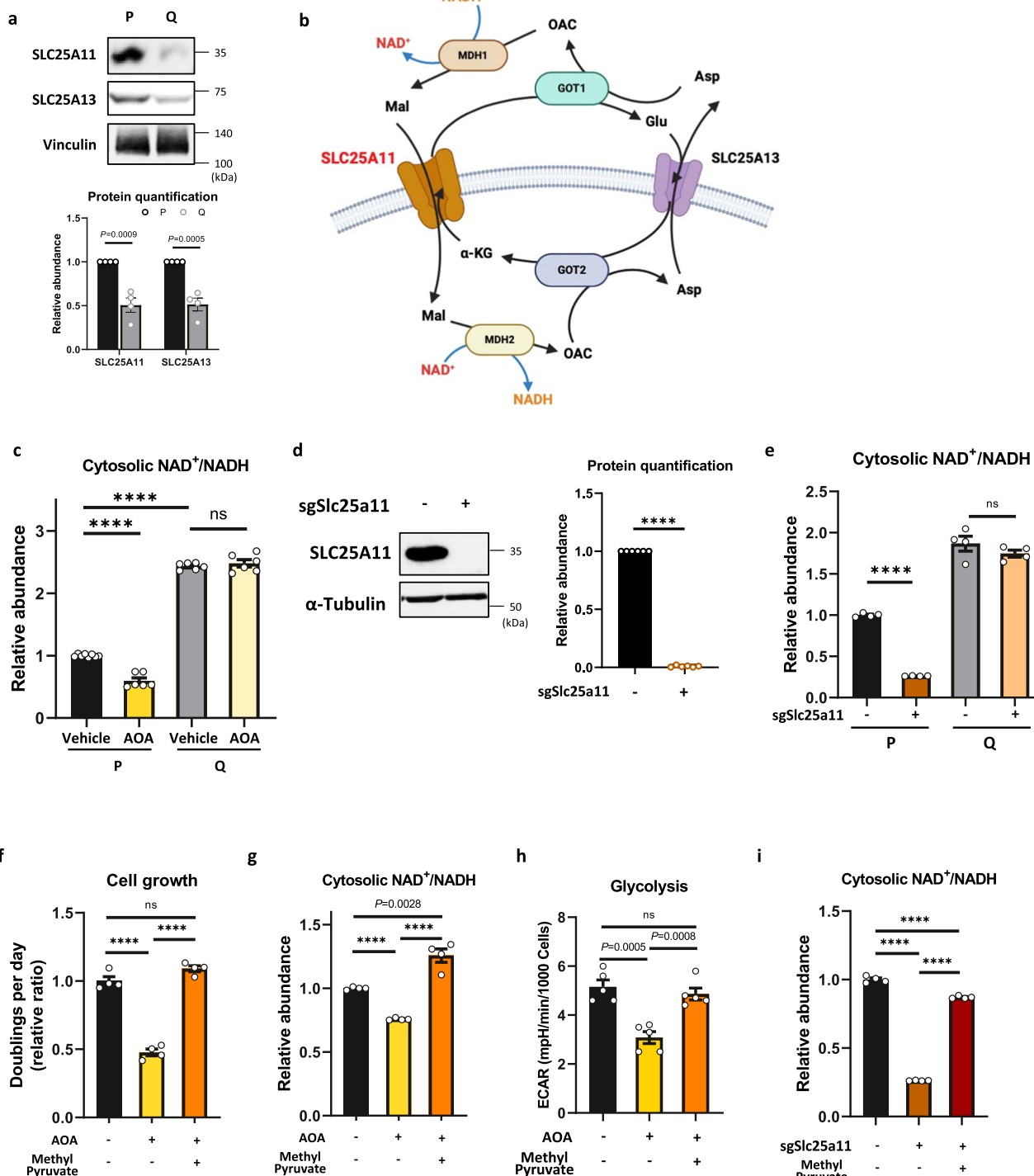

## RNA sequencing

A total of $3.0 \times 10^5$ cells were plated onto 6-well plates. After reaching proper growth states (P and Q), the cells were then rapidly washed two times in ice-cold PBS, and total RNA was isolated following the Direct-zol RNA Microprep Kit (ZYMO RESEARCH) manufacturer's manual. General quality-control metrics for next-generation sequencing data were obtained using FastQC. Sequencing trimming was performed using cutadapt[54]. Reads were aligned to the reference genome in a splice-aware manner using STAR[55], and apparent PCR duplicates were checked or removed with Picard. Reads were aligned to the reference genome using BWA MEM (arXiv:1303.3997). Afterward, the abundance of genomic features was quantified as raw counts based on read alignments using featureCounts[56]. The gene annotation database was Ensembl annotations for mouse genome v.mm39 (GRCm39).

## Gene set enrichment analysis (GSEA)

To gain insight into the transcriptional difference between P and Q cells, the GSEA tool was used to analyze the association between the P and Q profiles. In our study, the genomic expression profiles of 49,638 features (genes) were preranked using the log fold change as a ranking metric. A GSEA using the classic algorithm and the Reactome database was run with 3000 permutations.

**Fig. 7 | Quiescent cells suppress the malate-aspartate shuttle (MAS), whereas proliferating cells are dependent on it for cell proliferation. a** Immunoblots and protein quantification of SLC25A11 and SLC25A13 in P and Q MEFs. Vinculin was monitored as a loading control. Representative immunoblot is shown. Values are the mean ± SEM of 4 biologically independent experiments. *P* values by unpaired two-tailed student's *t*-test are indicated. **b** The schematic of malate-aspartate shuttle (MAS) (created with Biorender.com released under a Creative Commons Attribution-NonCommercial-NoDerivs 4.0 International license). MDH malate dehydrogenase, GOT glutamic-oxaloacetic transaminase, SLC25A11 solute carrier family 25 member 11. **c** The cytosolic NAD+/NADH ratio of P and Q MEFs cultured in vehicle or 2 mM AOA for 24 h measured by the pMOS023: Peredox NADH/NAD+ sensor (cytosolic) using flow cytometry. Values are the mean ± SEM of 6 biologically independent experiments. ****$P < 0.0001$ and ns $P = 0.4623$ using unpaired two-tailed *t*-test. **d** Immunoblot and protein quantification of SLC25A11 in MEFs. α-Tubulin was monitored as a loading control. Each immunoblot is representative of 6 biologically independent experiments. Values are the mean ± SEM of 6 biologically independent experiments; ****$P < 0.0001$ using unpaired two-tailed *t*-test. **e** The cytosolic NAD+/NADH ratio of P and Q MEFs with or without SLC25A11

deletion. The ratios are measured by the pMOS023: Peredox NADH/NAD+ sensor (cytosolic) using flow cytometry. Values are the mean ± SEM of 4 biologically independent experiments. ****$P < 0.0001$, ns $P = 0.2711$ using unpaired two-tailed *t*-test. **f** Proliferation rates of P MEF cells cultured in vehicle or 2 mM AOA or AOA + 4 mM methyl pyruvate for 24 h. The relative ratio of doublings per day is presented as the mean ± SEM of 4 biologically independent samples. ****$P < 0.0001$, ns $P = 0.0605$ using unpaired two-tailed *t*-test. **g** The cytosolic NAD+/NADH ratio of P cells cultured in vehicle or 2 mM AOA or AOA + 4 mM methyl pyruvate for 24 h. Data are presented as the mean ± SEM of 4 biologically independent samples. *P* value by unpaired two-tailed student's *t*-test is indicated except for ****$P < 0.0001$. **h** Glycolysis measured by the extracellular acidification rate (ECAR) in P cells cultured in vehicle or 2 mM AOA or AOA + 4 mM methyl pyruvate for 24 h. Data are presented as the mean ± SEM of 5 biologically independent samples. *P* values by unpaired two-tailed student's *t*-test are indicated except for ns $P = 0.4414$. **i** The cytosolic NAD+/NADH ratio of P MEFs with or without SLC25A11 deletion and cultured in vehicle or 4 mM methyl pyruvate for 24 h. Data are presented as the mean ± SEM of 4 biologically independent samples. ****$P < 0.0001$ using unpaired two-tailed *t*-test.

## Proliferation rates

Cells were plated in replicate 6-well plates in 3 mL at an initial seeding density of $3.0 \times 10^4$ cells per well. Cells were permitted to settle overnight. The number of cells seeded for each cell line allowed for exponential growth over the course of the assay. The following day, one 6-well plate of each condition was counted to determine the initial number of cells at the time of treatment. After 2 days of culture, final cell counts were measured. The cells were counted using a mixed solution of propidium iodide (PI) and Hoechst 33342 (Hoechst) with a final concentration of 2 µg/mL PI and 10 µg/ml Hoechst for 30 min at 37 °C. Plates were then imaged on a Celigo Image Cytometer (Nexcelcom Bioscience), and the number of PI-positive cells was divided by the total number of cells (Hoechst positive) to show the percentage of dead cells (% Dead). The following formula was used to calculate the proliferation rate: Doublings per day = [log2(final day 2 cell count/ initial day 0 cell count)]/2 days.

## Cell density and growth curve

The average cell density in a population was measured by a Celigo Image Cytometer (Nexcelcom Bioscience). Confluence (%) was monitored every 24 h for 13 days in a low using 'confluence' application with a hardware autofocus system. The mean ± SEM ($n = 3$) of confluence (%) of each population was plotted versus time (day) as a growth curve.

## Cell viability

Cells were plated on 12-well plates and allowed to grow until the P and Q states were reached. After reaching proper growth states (P and Q), the medium was aspirated and replenished with the indicated concentration of vehicle or UK5099. After 72 h, a mixed solution of propidium iodide (PI) and Hoechst 33342 (Hoechst) was added for a final concentration of 2 µg/mL PI and 10 µg/ml Hoechst for 30 min at 37 °C. Plates were then imaged on a Celigo Image Cytometer (Nexcelcom Bioscience), and the number of PI-positive cells was divided by the total number of cells (Hoechst positive) to show the percentage of dead cells (% Dead).

## Extracellular acidification rate (ECAR) and oxygen consumption rate (OCR)

The extracellular acidification rate (ECAR) and oxygen consumption rate (OCR) were measured using an Agilent Seahorse XFe96 Analyzer using standard methods. Briefly, cells were plated at $1.5 \times 10^3$ cells per well in Seahorse Bioscience 96-well plates. After settling, the cells were washed three times and incubated in Seahorse XF base medium (Agilent, 102353-100) pH adjusted to 7.4 with the indicated treatment. The assay medium for ECAR measurements contained glutamine (4 mM;

Sigma), and the following compounds were injected: glucose (10 mM; Sigma–Aldrich), oligomycin A (2 µM), and 2DG (100 mM; Sigma–Aldrich). This allowed for calculation of the glycolytic rate, glycolytic capacity, and glycolytic reserve. The basal ECAR was measured prior to the addition of glucose. The assay medium for OCR measurements contained glucose (25 mM; Sigma) and glutamine (4 mM; Sigma). The following inhibitors were injected for the OCR: oligomycin A (2 µM; Cayman Chemical, Ann Arbor, MI, USA), carbonyl cyanide 4-(trifluoromethoxy) phenylhydrazone (FCCP) (2 µM; Sigma–Aldrich), antimycin A (1 µM; Sigma–Aldrich) and rotenone (1 µM; Sigma–Aldrich). This method allowed for the calculation of the OCR-linked ATP production, maximal respiratory capacity, and spare respiratory capacity. Basal respiration was measured prior to the addition of oligomycin A.

## Measurement of glucose consumption

Medium was collected from cells cultured in vehicle or 0.2 µM rotenone for 24 h. The glucose concentrations were measured on a YSI-2700 Biochemistry Analyzer (Yellow Springs Instruments) as described previously[57], and the rate of glucose consumption from the media was normalized to the cell counts.

## Metabolite measurement by GC-MS

For metabolite measurement by isotope labeling, $3.0 \times 10^5$ cells were plated onto 6-well plates. After reaching proper growth states (P and Q), isotope labeling experiments were performed for 24 h with either 25 mM [1,2-13C] glucose (Cambridge Isotope Laboratories), 25 mM [U-13C] glucose, or 4 mM [U-13C] glutamine, and extra plates were included for cell counts. At the time of collection, the plates were washed twice with 1 mL of (9 g/L) NaCl. Then, the cells were incubated with 600 µL of mixed solvent (water:methanol:acetonitrile = 1:1:1) containing 2 µL of 2 mg/mL Norvaline (Sigma–Aldrich) dissolved in distilled water as an internal standard and then scraped down with cell scrapers. The solution was shaken at $500 \times g$ for 30 min and centrifuged at $16,000 \times g$ for 15 min at 4 °C. The supernatant (280 µL) was transferred to a clean tube and lyophilized under nitrogen gas. For the analysis of DHAP, pyruvate, lactate and citrate, the lyophilized samples were derivatized with 15 µL of 2 wt% methoxyamine hydrochloride (Thermo Fisher) for 60 min at 42 °C. Next, 35 µL of N-methyl-N-(tert-butyldimethylsilyl)-trifluoroacetamide (MTBSTFA) + 1% tert-butyldimetheylchlorosilane (t-BDMCS) (Sigma–Aldrich) was added, and the samples were incubated for 70 min at 75 °C. For the analysis of G6P, the lyophilized samples were derivatized with 20 µL of 2 wt% methoxyamine hydrochloride (Thermo Fisher) for 60 min at 42 °C. Next, 20 µL of N-methyl-N-(trimethylsilyl)trifluoroacetamide (MSTFA) (Sigma–Aldrich) was added, and the samples were incubated for 30 min at 37 °C. The derivatized samples were then centrifuged at

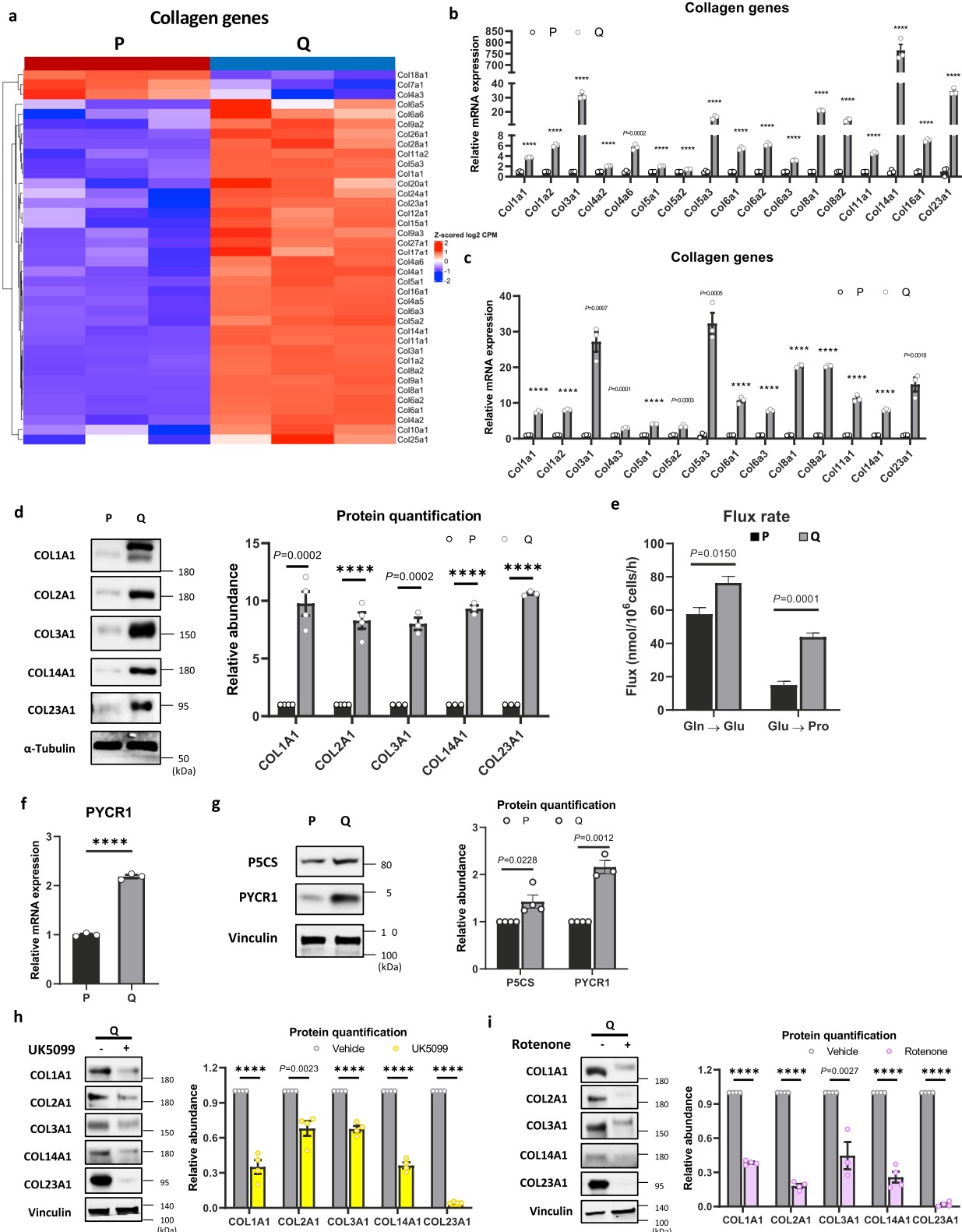

16,000 × *g* for 5 min at 4 °C, and the supernatant (1 μL) was subjected to GC–MS measurement.

GC–MS analysis was performed using an Agilent 7890B GC equipped with a DB-5ms column (30 m × 0.25-mm inner diameter; film thickness, 0.25 μm; Agilent J&W Scientific). The front inlet temperature was 280 °C, and the helium flow was maintained at 1.428 mL/min. For DHAP, pyruvate, lactate, and citrate analysis, the column temperature

was held at 60 °C for 1 min followed by 1 min of run time, with an increase of 10 °C/min to 320 °C and holding for 1 min followed by a postrun of 320 °C for 9 min. For the analysis of G6P, the initial rate was held at 60 °C for 1 min followed by 1 min of run time, with an increase of 10 °C/min to 325 °C and holding for 10 min followed by a postrun of 60 °C for 1 min[58,59]. The peak area of each quantified ion was calculated and corrected for natural isotope abundances by following the

**Fig. 8 | Active mitochondrial metabolism supports ECM biosynthesis in quiescent cells. a** Heatmap of the 38 genes encoding collagen between P and Q MEFs as determined by RNA sequencing. Data represent values from three independent experiments, and Z-scored log2-fold change values are color-coded as indicated. **b** Relative mRNA expression of collagen genes quantified by RNA sequencing in P and Q MEFs. Means ± SEMs (n = 3) are shown. P value by unpaired two-tailed student's t-test is indicated except for ****P < 0.0001. **c** Relative mRNA expression of collagen genes quantified by RT-qPCR in P and Q MEFs. Means ± SEMs (n = 3) are shown. P value by unpaired two-tailed student's t-test is indicated except for ****P < 0.0001. **d** Immunoblots and protein quantification of Collagen Iα1 (COL1A1), Collagen IIα1 (COL2A1), Collagen IIIα1 (COL3A1), and Collagen XXIIIα1 (COL23A1) in P and Q MEFs. Representative immunoblots are shown. Values are the mean ± SEM of 3 biologically independent experiments. P value by unpaired two-tailed student's t-test is indicated except for ****P < 0.0001. **e** Comparison of the quantified intracellular metabolic fluxes (estimated flux ± SD) of Proline synthesis (Gln → Glu and Glu → Pro) in P and Q MEFs. All flux fits passed the chi-square goodness-of-fit test. P values by unpaired two-tailed student's t-test are indicated.

**f** Relative mRNA expression of pyrroline-5-carboxylate reductase 1 (PYCR1) quantified by RNA sequencing in P and Q MEFs. Means ± SEMs (n = 3) are shown. ****P < 0.0001 using unpaired two-tailed t-test. **g** Immunoblots and protein quantification of P5CS and PYCR1 in P and Q MEFs. Representative immunoblots are shown. Values are the mean ± SEM of 3–4 biologically independent experiments. P values by unpaired two-tailed student's t-test are indicated. **h** Immunoblots and protein quantification of Collagen Iα1 (COL1A1), Collagen IIα1 (COL2A1), Collagen IIIα1 (COL3A1), Collagen XIVα1 (COL14A1), and Collagen XXIIIα1 (COL23A1) in Q MEFs cultured in vehicle or 100 μM UK5099 for 24 h. Representative immunoblots are shown. Values are the mean ± SEM of 4 biologically independent experiments. P value by unpaired two-tailed student's t-test is indicated except for ****P < 0.0001. **i** Immunoblots and protein quantification of Collagen Iα1 (COL1A1), Collagen IIα1 (COL2A1), Collagen IIIα1 (COL3A1), Collagen XIVα1 (COL14A1), and Collagen XXIIIα1 (COL23A1) in Q MEFs cultured in vehicle or 0.2 μM rotenone for 24 h. Representative immunoblots are shown. Values are the mean ± SEM of 3 or 4 biologically independent experiments. P value by unpaired two-tailed student's t-test is indicated except for **** P < 0.0001.

procedure reported by Fernandez et al.[60] and then normalized by the peak area of norvaline as an internal standard and cell counts.

## $^{13}$C-Metabolic flux analysis ($^{13}$C-MFA)

$3.0 \times 10^5$ cells were plated onto 6-well plates. After reaching proper growth states (P and Q), isotope parallel labeling experiments were performed for 0, 6, 18, and 24 h timepoints with either 25 mM [1,2-$^{13}$C] glucose (Cambridge Isotope Laboratories), or 4 mM [U-$^{13}$C] glutamine (Cambridge Isotope Laboratories). Extra plates were included for cell counts for all time points, including the starting point. For each of the labeled plates, 200 μL of sample media was collected for extracellular metabolites. For intracellular metabolites, plates were washed twice with 1 mL of (9 g/L) NaCl. Then, the cells were incubated with 1.5 mL of ice-cold methanol for 5 min, scraped down with cell scrapers, and transferred into glass centrifuge tubes (Kimble Chase). Next, 1.5 mL of chloroform was added to the tubes and vortexed at high speed for 10 s. Then, 1.5 mL of HPLC grade water was added, and the tubes were vortexed at high speed for 1 min, capped, and stored at 4 °C overnight. The next day, the samples were centrifuged at $650 \times g$ for 20 min at 4 °C. Then, 3 mL of the aqueous phase was transferred to two 1.5-mL centrifuge tubes, dried completely under nitrogen gas at 37 °C, and stored at −80 °C. Finally, 1.5 mL of the organic phase was transferred to new glass centrifuge tubes and stored at −80 °C. Samples were then subjected to GC/MS analysis.

## Determination of external rates for $^{13}$C-MFA

To quantify concentrations of metabolites in spent media, samples were collected at 0, 6, 18, and 24 h and mixed with an internal standard solution containing $^{13}$C-labeled metabolites with a known concentration. The samples were then derivatized using MOX-TBDMS and analyzed by GC-MS as described before[61]. Metabolite uptake and secretion rates were then calculated as described by Antoniewicz[62], using equations 4 and 5 in this reference for proliferating and quiescent cells, respectively.

## Measurement of isotopic labeling for $^{13}$C-MFA

For GC-MS analysis of isotopic labeling of intracellular metabolites, dried cell extracts were derivatized using the MOX-TBDMS derivatization method as described by Oates and Antoniewicz[61]. The identity of the measured metabolites was verified using pure analytical standards as described by Long and Antoniewicz[13]. For GC-MS analysis of isotopic labeling of ribose from RNA, the protocol described by Long et al.[63] was used. In all cases, GC-MS analysis was performed on an Agilent 7890 A GC system equipped with a HP-5MS capillary column (30 m, 0.25 mm i.d., 0.25μm-phase thickness; Agilent J&W Scientific), connected to an Agilent 5977B Mass Spectrometer operating under ionization by electron impact (EI) at 70 eV. Helium flow was maintained at 1 mL/min. The source temperature was maintained at 230 °C, the MS

quad temperature at 150 °C, the interface temperature at 280 °C, and the inlet temperature at 280 °C. Mass spectra were recorded in single ion monitoring (SIM) mode with 4 ms dwell time on each ion. Mass isotopomer distributions were obtained by integration of ion chromatograms[64] and corrected for natural isotope abundances[60].

Intracellular metabolic fluxes were determined by fitting extracellular uptake and secretion rates (glucose, glutamine, lactate, alanine, pyruvate, glutamate, and proline) and the measured mass isotopomer distributions of intracellular metabolites (pyruvate, lactate, alanine, malate, aspartate, α-ketoglutarate, glutamine, citrate, glycerol 3-phosphate, phosphoenolpyruvate, 3-phosphoglycerate, and ribose from RNA) to a compartmentalized metabolic network model using the Metran software[13]. The model contains three distinct metabolic compartments: extracellular, cytosol, and mitochondrion. The metabolites pyruvate, acetyl-CoA, citrate, α-ketoglutarate, malate, oxaloacetate, alanine, glutamate, and aspartate are metabolically active in both the cytosol and mitochondrion. During the extraction process, intracellular pools of metabolites are homogenized. As such, the measured isotopic labeling of these metabolites reflects the mixture of distinct metabolic pools. In the $^{13}$C-MFA model, we included mixing reactions to account for the mixing of mitochondrial and cytosolic metabolite pools during extraction[65]. The model also accounts for dilution of intracellular metabolites due to incorporation of unlabeled $CO_2$ and influx of unlabeled metabolites from the medium such as glutamate, aspartate, and pyruvate.

For $^{13}$C-MFA, data from four parallel labeling experiments for each condition, two biological replicates using [1,2-$^{13}$C] glucose as tracer and two biological replicates using [U-$^{13}$C] glutamine as tracer (samples collected 24 h after tracer addition), were fitted simultaneously to the network model to estimate intracellular fluxes. To ensure that the global best solution was identified, flux estimation was repeated at least 50 times starting with random initial values. At convergence, a chi-square test was applied to test the goodness-of-fit, and accurate 95% confidence intervals were calculated by determining the sensitivity of the sum of squared residuals to flux parameter variations[66].

## Coimmunoprecipitation (Co-IP) and immunoblotting

For coimmunoprecipitation (Co-IP), cells grown in 6-well plates were lysed in RIPA lysis buffer (Cell Signaling) with a protease inhibitor mini tablet (Thermo Fisher), and a phosphatase inhibitor mini tablet (Thermo Fisher, 1 tablet per 10 ml of lysis buffer). Solubilized proteins were collected by centrifugation and quantified using protein assay reagent (Bio-Rad). Protein extracts were incubated with each primary antibody with rotation overnight and then were added to protein A/G agarose beads (Santa Cruz, sc-2003) for an additional 1 h. The immunoprecipitates were washed three times with RIPA buffer and then resuspended in sample buffer for immunoblotting.

For immunoblotting, protein extracts were prepared in lysis buffer as described previously[67]. Solubilized proteins were collected by centrifugation and quantified using protein assay reagent (Bio-Rad). Samples containing equal amounts of protein were resolved by electrophoresis on a 6–15% gel, and the proteins were transferred to polyvinylidene difluoride (PVDF) membranes (Millipore). Primary antibody incubation was performed in 2.5% milk or 2.5% BSA in TBS-T, and secondary antibody incubation was performed in 2.5% milk. Standard enhanced chemiluminescence (ECL) was used, and an Azure or LI-COR machine was used to image the blots. ImageStudio software (Licor) was used to quantify the bands on the blots.

### RT–qPCR

RNA was isolated with a GeneJET RNA Purification Kit (Thermo Fisher K0732) according to the manufacturer's protocol. cDNA was synthesized with an iScript cDNA synthesis kit (Bio-Rad 1708891), and RT–qPCR was performed with Bio-Rad iQ SYBR Green Supermix (Bio-Rad 1708882) according to the manufacturer's protocol on a Bio-Rad CFX96. Vinculin or cyclophilin E was used as a reference gene. RT–qPCR primer sequences are provided in Supplementary Table 1. Values are recorded as the fold change calculated by the ddCT method.

### Immunofluorescence

Cells were plated in triplicate in 8-well culture slides. Cells were fixed in 4% paraformaldehyde (PFA) for 30 min followed by four washes in ice-cold PBS. After fixation of the cells, the localization of YAP (green) and nuclei (blue; Hoechst 33342) was examined under a confocal microscope (Zeiss LSM 700 or LSM 900) in at least 10 randomly selected fields at 400× magnification.

### 8XGTIIC (TEAD) luciferase reporter assay

A total of $3.0 \times 10^5$ cells were plated onto 6-well plates. After reaching proper growth states (P and Q), the cells were then cotransfected with 8xGTIIC-luciferase and Renilla construct using FuGENE (Promega) following the manufacturer's instructions. 48 h post-transfection, the cells were lysed, and firefly/Renilla luciferase activity was quantified by luminescence using a Dual-Luciferase® Reporter Assay System (Promega) following the manufacturer's instructions.

### Bis(sulfosuccinimidyl)suberate (BS³) cross-linking analysis

Cross-linking analysis was performed with BS³ Crosslinkers (Thermo) following the manufacturer's instructions. Briefly, cells were suspended and washed three times with ice-cold PBS (pH 8.0) to remove amine-containing culture media and proteins from the cells. Then, 2 mM BS³ solution was added, and the reaction mixture was incubated for 30 min at room temperature. Next, quenching solution was added to a final concentration of 10–20 mM Tris and incubated for 15 min at room temperature. The samples were then lysed and prepared for electrophoresis.

### Bromodeoxyuridine (BrdU) and Propidium Iodide (PI) staining for cell cycle

A total of $3.0 \times 10^5$ cells were plated onto 6-well plates. After reaching proper growth states (P and Q), the cells were then incubated with 3 ug/mL BrDU for 3 h, and then were harvested and fixed. Cells were denaturized in 2 N HCl + 0.5% Triton X and washed, then the residual acid was neutralized with 0.1 M sodium borate pH8.5. Cells were incubated with primary monoclonal antibody, Ms anti-Bromodeoxyuridine Clone Bu20a (Dako M0744), overnight. After washed, cells were incubated in anti-Ms Alexa 488 for 2 h protected from light. The pellets were resuspended in PI solution with RNase A for 30 min protected from light. Cells were washed with PBS, and resuspended in FACS buffer (2% FBS, 2 mM EDTA). Alexa 488 for BrdU and PE for PI fluorescences were measured on a Beckman Coulter CytoFLEX S flow cytometer.

### Mitochondrial membrane potential ($\Delta\Psi_m$) measurement

Mitochondrial membrane potential ($\Delta\Psi_m$) was assessed using TMRE (tetramethylrhodamine, ethyl ester) (Invitrogen, T669). The intensity of TMRE was determined as described previously[68]. Cells were plated onto 12-well plates at a plating density of $3.0 \times 10^4$ cells per well. After reaching proper growth states (P and Q), the cells were then treated with 100 nM TMRE for 30 min, trypsinized, washed with PBS, and resuspended in FACS buffer (2% FBS, 2 mM EDTA). TMRE fluorescence was measured on a Beckman Coulter CytoFLEX S flow cytometer.

### Whole-cell NAD⁺/NADH ratio measurement

A total of $3.0 \times 10^5$ cells were plated onto 6-well plates. After reaching proper growth states (P and Q), the cells were then rapidly washed two times in ice-cold PBS and extracted in 100 mL of ice-cold lysis buffer (1% dodecyltrimethylammonium bromide [DTAB] in 0.2 N NaOH diluted 1:1 with PBS). The NAD⁺/NADH ratio was measured using an NAD/NADH-Glo Assay kit (Promega, G9072) according to a modified protocol as described previously[69]. Briefly, to measure NAD⁺, the lysate was diluted with lysis buffer and 0.4 N HCl and subsequently incubated at 60 °C for 15 min. For NADH measurement, the lysate was incubated at 75 °C for 30 min. Acidic conditions permit the selective degradation of NADH, while basic conditions degrade NAD⁺. Following incubation, the samples were spun on a bench-top centrifuge and quenched with neutralizing solution. The neutralizing solution consisted of 0.5 M Tris base for NAD⁺ samples and 0.25 M Tris in 0.2 N HCl for the NADH samples. The instructions in the Promega G9072 technical manual were then followed to measure NAD⁺ and NADH levels using a luminometer (BioTek).

### Cytosolic and mitochondrial NAD⁺/NADH ratio measurement

For cytosolic NAD⁺/NADH ratio measurement with the Peredox probe, cells were transduced with the pMOS023: Peredox NADH/NAD⁺ sensor (cytosolic) plasmid (Addgene 163060). For mitochondrial NADH measurement, cells were transfected with pC1-mitoRexYFP (Addgene 60246). Cells were plated onto 12-well plates at a plating density of $3.0 \times 10^4$ cells per well. After reaching proper growth states (P and Q), the cells were trypsinized and rapidly washed two times in ice-cold PBS. Cells were analyzed on a CytoFlex flow cytometer. Data analysis was performed using FlowJo software. Data were reported as the mean fluorescence intensity (MFI) normalized to the untransduced negative control sample.

### Cytosolic NADP⁺/NADPH ratio measurement with the iNap1 sensor

For cytosolic NADP⁺, NADPH, and NADP⁺/NADPH ratio measurement with the iNap1 sensor, cells were transduced with the pPB iNap1 sensor (cytosolic) plasmid[40]. Cells were plated onto 12-well plates at a plating density of $3.0 \times 10^4$ cells per well. After reaching proper growth states (P and Q), the cells were trypsinized and rapidly washed two times in ice-cold PBS. Cells were analyzed on a CytoFlex flow cytometer. iNap1 signals were acquired with excitation wavelengths of 405 nm and 488 nm and an emission wavelength of 525 nm. Data analysis was performed using FlowJo software. Data are reported as the MFI normalized to the untransduced negative control sample.

### Quantification and statistical analysis

Statistical analysis was performed with built-in statistics tools in Prism9 (GraphPad Software). Statistical significance was determined using unpaired two-tailed Student's t-test and ordinary one-way analysis of variance (ANOVA) with Dunnett's or Holm–Sidak's multiple comparison test or two-way ANOVA with Tukey's post hoc multiple comparison test in comparisons of more than two groups of samples. Welch's unequal variances t-test used for the flux analyses. Statistical tests and error bars can be found in the figure legends. *P* values for comparisons

between groups are indicated in the figures. The statistical method used, and sample size (*n*) are indicated in the legends for each figure.

### Reporting summary

Further information on research design is available in the Nature Portfolio Reporting Summary linked to this article.

## Data availability

Data are available within the Article, Supplementary data tables, Supplementary Information or Source Data file. The RNA-seq data reported in this study has been deposited in the NCBI Gene Expression Omnibus (GEO) under accession number GSE272680. Source data are provided with this paper.

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

## Acknowledgements

N.H. acknowledges support from NIH grants R01AG016927, R01CA090764, R01CA206167, and R01CA258299; the VA merit awards BX000733 and BX005092; and the VA research career scientist award IK6BX004602. We thank UIC Genome Research Core and Research Informatics Core for their help with next-generation sequencing and RNA sequencing experiments. Figs. 5k and 7b were created with BioRender.com released under a Creative Commons Attribution-NonCommercial-NoDerivs 4.0 International license (https://creativecommons.org/licenses/by-nc-nd/4.0/deed.en).

## Author contributions

N.H. and S.K. designed the study. S.K. conducted the experiments and M.R.A. designed the metabolic flux analyses.

## Competing interests

The authors declare no competing interests.
