## [Peer Review File · Nature Communications]

REVIEWER COMMENTS

Reviewer #1 (Remarks to the Author):

This manuscript, “Metabolic and transcriptomic reprogramming in the transition from a proliferative to quiescent state is mediated by YAP-dependent and Yap-independent mechanisms” reports an extensive descriptive study of the metabolic rewiring and transcriptional changes that occur as serum-stimulated fibroblasts transition from a subconfluent, proliferating phenotype to a nonproliferating contact-inhibited phenotype. Upon contact inhibition, fibroblasts exhibit reduced glycolysis and increased TCA cycle activity. There is also a shift from subconfluent cell proliferation initiated by serum-dependent growth factors to a phenotype where contact-inhibited fibroblasts in the presence of 10% serum begin to adopt an effector phenotype characterized by de novo extracellular matrix protein synthesis and secretion. As previously reported by others, this transition is mediated in part by E-cadherin inhibition of YAP nuclear translocation and inhibition of TEAD-dependent transcription (ref. 26, 27). While it has been widely reported that contact-inhibited fibroblasts exhibit a high metabolic activity and secrete increased levels of matrix proteins (ref. 7, 12, 14), there is yet to be a comprehensive study of how either transcription or metabolism is reprogrammed as fibroblasts transition from a proliferative phenotype to an effector phenotype. As such, the data presented in the manuscript provide a comprehensive profile of the metabolic and transcriptional changes in this well studied model of cell proliferation arrest.

Despite the comprehensive nature of this study, it is disappointing that only one of the transcriptional programs thought to regulate this transition is studied in detail. While the Yap/TEAD transcriptional arm is investigated, The Yap-independent mechanisms mentioned in the title are not further investigated. Previous work has suggested that β -catenin and p53 also play a role in the reprogramming during this transition and it is disappointing that more effort was not made to characterize the contributions of these transcriptional mediators. Although the authors study both MEFs immortalized with dominant-negative p53 as well as wildtype MEFs, a detailed characterization of the metabolic and transcriptional differences between these two cell types is not provided and the authors only report the similarities between cells with or without p53 transcriptional activation.

Overall, the manuscript reports valuable information concerning this well characterized model in which proliferating fibroblasts transitions between serum-induced proliferative function and serum-dependent effector function(also called senescence by some). The data will be of use to others involved in this well-studied cell model. Before publication, the following issues should be addressed:

1. The authors should provide data concerning the transcriptional and metabolic differences between their wildtype MEFs and the MEFs carrying a p53 dominant-negative used in these studies. p53 has been established as a key transcriptional regulator of metabolic genes and how much of the Yap-independent mechanisms reported in the manuscript can be accounted for by p53-dependent changes in transcription should be determined since these reagents are on hand.

2)The introduction and discussion should be revised to make it clear that the data reported apply to only one well-characterized model of embryonic fibroblasts quiescence, that of contact-inhibition. The other major model of embryonic fibroblasts quiescence that is also widely studied is that of serum-deprivation. While many past studies have utilized both models to study non-proliferating fibroblasts, there are clear differences between phenotypes of fibroblasts that have ceased proliferating following serum starvation and those that arrest through contact-inhibition in the presence of serum growth factors. This should be clarified for the general reader.

Reviewer #2 (Remarks to the Author):

Kang et al. use metabolic flux analysis in mouse embryonic fibroblasts (MEFs) to investigate the metabolic differences of proliferating and quiescent cells. They find extensive metabolic rewiring during the transition from the proliferative towards the quiescent state, most prominently a reduction in glycolytic flux with a concomitant increase in TCA cycle flux in quiescent cells. The authors went on to show that increased E-Cadherin expression in quiescent cells suppresses glycolytic flux via inhibition of Yap/TAZ activity. Mediators of increased TCA cycle flux may involve differential expression of the mitochondrial pyruvate carrier. Finally, the authors propose that the metabolic rewiring observed in quiescent fibroblasts supports extracellular matrix synthesis.

The authors provide a fresh perspective on how cellular metabolism supports the metabolic requirements of proliferative and non-proliferative cells and thereby address a gap in the field. Several of their findings are novel and should attract interest in the field, including the inhibition of glycolysis via E-Cadherin mediated suppression of YAP/TAZ, and the impaired dimerization of ME1, in the quiescent cells. The metabolic experiments are technically sound and support most of their conclusions. However, there are several major issues that I list below and suggest the authors to address before consideration for publication. While some mechanistic studies on the underpinnings of the metabolic phenotypes have been performed, they are incomplete. In addition, some important literature is not cited. Overall, the study falls short of connecting the observed metabolic differences to any functional consequences, which reduces the significance of the reported findings. Lastly, the relevance of the findings beyond fibroblasts remains unclear.

1. The authors need to provide more details on the ¹³C-MFA experiment in the methods as this is the key experiment of the study. For example, they state that the cells were plated and pulse-labeled the next day, which indicates they haven't been subjected to the same proliferation/quiescence protocol as described in Figure 1A. Have they been treated like this and then replated for the tracer studies? In addition, it remains unclear which of the time points for the tracing described was used to derive the model. The authors should also describe how they verified the nature of the metabolites measured.

2. A similar problem exists for the extracellular flux analysis - how were the cells brought into the P and Q states for seahorse assays?

3. The authors suggest higher MPC expression promotes the increased pyruvate flux into mitochondria. How is MPC expression regulated in quiescent vs proliferative cells?

4. Some of the transcriptomic changes observed by the authors in the quiescent cells seem to be specific to fibroblasts, including those related to the ECM. To broaden the significance of their study, the authors should perform some of their key experiments in other relevant cell types, comparing the proliferative and quiescent state.

5. The authors use AOA as a MAS inhibitor. However, AOA broadly inhibits many transaminases and is not specific to the MAS. Therefore, to support their conclusions, the authors need to provide more evidence, for example using a genetic approach. This may include deletion of *Slc25a11* which is relevant in their experimental conditions as it is downregulated in Q compared to P cells.

6. The authors fail to cite important primary literature regarding the role of proline metabolism for collagen synthesis in fibroblasts (PMID: 30973753, 32134147, 35760868). Moreover, *PYCR1* is not the key mitochondrial enzyme responsible for proline biosynthesis, but *P5CS / Aldh18a1* is the rate-limiting enzyme, as shown in embryonic fibroblasts like those used in this study (PMID: 32134147).

7. The title of the last section needs to be corrected to “Active mitochondrial metabolism may support ECM biosynthesis in quiescent cells”, as no experimental evidence supporting this conclusion is provided.

8. To support the significance of their findings, the authors should provide mechanistic evidence that the metabolic rewiring in the quiescent cells is functionally relevant. Are some of the observed changes required to maintain the quiescent state? Does the metabolic rewiring indeed support ECM production as suggested by the authors?

9. It would also be helpful if the authors could provide a mechanism by which the increased MPC expression or higher flux of glutamine into proline is regulated in the quiescent cells.

Minor:

1. The authors should cut down on their introduction - the end is repetitive with the discussion.

2. Page 5, the authors state that “most TCA cycle enzymes ... were significantly upregulated in Q cells”, but they only look at four, which is not most.

3. Page 6, end of first paragraph, the authors suggest that “attenuated glycolysis and induced mitochondrial metabolism are global phenomena in quiescent fibroblasts”, but they only ever look at metabolic enzyme expression in the other fibroblasts investigated, which is not the same as glycolysis or mitochondrial metabolism.

4. Page 6, the authors suggest that Q E-Cad KO cells are more energetic, but no experimental data to support a higher bioenergetic state are provided.

5. Page 7, the authors need to show successful deletion and re-expression of wt and mutant YAP in their cells, e.g. by western blot.

6. Page 9, based on Fig. 6E the authors state that ACAT1-mediated acetylation of ME1 was higher in P cells than in Q cells, but the experiment only shows a correlation between acetylation and ACAT1, not that ACAT1 mediates the acetylation of ME1 under their experimental conditions

7. Page 10, the authors state that the only source of NADPH in Q cells is the PPP, but they do not show that. For the cytosolic part, they do not consider IDH1. In the statement as is, it also is not specific to the cytosol, which is what I believe the authors are trying to say. Moreover, if the Q cells don't require high NADPH as suggested by the authors, shouldn't they be less sensitive to inhibition of the PPP?

Reviewer #3 (Remarks to the Author):

In this manuscript, Kang S and colleagues investigate the metabolic modifications occurring during the transition from a proliferative to a quiescent state in mouse embryonic fibroblasts. They combine metabolic flux and transcriptomic analysis to identify the metabolic pathways that are the mostly modulated upon cellular switch from proliferation to quiescence.

The authors nicely demonstrate that quiescent cells undergo a reduction in glucose uptake and glycolytic flux, which is mostly mediated by a reduction in the expression of glycolytic genes when compared to proliferative cells. This quiescence-induced reduction of glycolysis is largely dependent on increased E cadherin expression and consequently reduced YAP activity. While glycolysis is inhibited, the flux of pyruvate into mitochondria and the oxidation acetyl-CoA in the TCA cycle and OXPHOS pathways are markedly increased in quiescent cells in YAP-independent manner, and an increased expression of MDH1 supports leading to a global metabolic rewiring that stimulates the conversion of cytosolic malate to oxaloacetate, PEP and pyruvate.

The study is rigorously conducted and provides a global view of the main metabolic changes occurring in quiescent immortalized fibroblasts when compared to proliferating ones. While confirming previously published data (i.e., that non proliferating cells switch from glycolysis to mitochondrial oxidation of pyruvate and acetyl-CoA), this study reveals new details about the general rewiring of crucial metabolic pathways, as well as of NAD⁺ and NAD⁺/NADH ratio, in quiescent cells when compared to proliferating ones.

The manuscript is written in a very clear way. The experimental approach is also appropriate, and individual experiments are interpreted in the correct way.

Therefore, in my opinion this manuscript could be considered for publication in Nature Communications.

However, I have major concerns that must be addressed before further considering this work for publication.

Major issues

- The main limitation of this study is the fact that the quiescent state of fibroblasts is achieved by allowing cells reach confluence in the growth plate. Therefore, it is difficult to understand what metabolic changes occurring in quiescent cells are the results of quiescence itself, and what metabolic adaptive modifications are the result of the fact that cells reach 100% confluence in the plate, which comes with a significant reduction of glucose and amino acid concentration in cell growth media. For instance, while an increased expression of E cadherin is clearly responsible for reduced YAP expression and lowered glycolytic flux in quiescent cells, it is unclear whether the observed increased in mitochondrial oxidation of pyruvate and acetyl-CoA depends on E-cadherin-induced quiescence or, alternatively, on cellular confluence and consequent shortage of extracellular glucose and other metabolites. Similarly, it is unclear whether an increased expression of mitochondrial transporters of pyruvate is essential for the survival of quiescent cells because of changes in their proliferative state, or because in the experimental system used in this work quiescence reflects a drop in extracellular metabolites, which forces tumor cells to enhance

mitochondrial pyruvate oxidation to survive. To address this point, I suggest to replicate the major study findings by also using an alternative experimental approach to induce cellular quiescence

- How many times did the authors change the growth medium in quiescence cells? The authors state that the quiescent state is reached because of contact-mediated inhibition of fibroblast proliferation. However, fibroblasts may also stop proliferating because of the shortage of extracellular metabolites, such as glucose or amino acids, in the growth medium of confluent cells after several days of growth. To circumvent this major limitation of the experimental system used in this work, I suggest to repeat at least the major experiments by refreshing medium in order to exclude the possibility that the reduced glycolytic flux should be dependent on the absence of glucose in the growth medium 11 days after seeding.

- I disagree with one of the main study conclusions, i.e., that the metabolic switch occurring during the transition into a quiescent state is a potentially universal mechanism that may occur in different cell types, including normal epithelial cells or even cancer cells. Indeed, different cell types can be highly heterogeneous in terms of metabolic requirements and metabolic responses to external stress or changes in their microenvironment. In this work, the authors only used a single cell line, which was represented mouse embryonic fibroblasts (MEFs), and they just confirmed that some of the TCA cycle enzymes are increased on another murine quiescent cell line, NIH 3T3, and in human BJ fibroblasts. I recommend to replicate the main study findings in at least another model of non-tumoral cells, and in at least one model of cancer cells.

- Why did the authors not evaluate the expression of a lot of glycolytic enzymes which were studied in the case of MEFs (Fig 1 K)? Why did they just evaluate the expression of HK2 and PFK1 in the case of NIH 3T3 and BJ cells? I suggest to show the expression of the other glycolytic enzymes in the two additional models;

- What is the impact of YAP overexpression on quiescence-induced metabolic rewiring?

- Which is the correlation between YAP constitutively activation and an increment in TCA cycle in quiescent cells? I suggest to demonstrate with ^{13}C glucose flux that that an increased glycolysis, caused by YAP activation, could in turn induce an even higher activation of TCA cycle;

- Since it is well known from the literature that YAP is implicated in cell migration, I suggest to the authors to evaluate the impact of YAP overexpression or KO on MEF cell migration by using a wound healing assay in non-confluent (non-quiescent) cells

- I disagree with the interpretation of the authors about the reason why ME1 flux is suppressed in quiescent cells. Indeed, in order to prove that dimerization of ME1, and consequently also its activity, is reduced because of an activation of PPP, which require to quiescent cells the proper amount of NADPH, a KO of some genes involved in the PPP pathway is required. If the hypothesis is correct, ME1 dimerization should be observed. I suggest to KO some of the limiting enzyme of PPP pathway, like Glucose-6-phosphate dehydrogenase;

- The authors clarify very well the role of YAP in metabolic reprogramming by reducing glycolytic flux in MEFs quiescent cells. Anyway, they did not rescue the quiescent state of cells with constitutively active form of YAP, which let them to conclude that quiescent state of cells is independent on YAP. Have you considered the idea to use a different type of model? If the model is characterized by cells

which completely covered the plate and achieve 100% confluence, how did the authors expect a rescue of quiescence? I suggest to repeat the experiment by inducing cells quiescence through serum starvation;

- Did the authors have check the expression of senescence markers, such as p21 or b-galactosidase, in quiescent cells? ;

- The authors indirectly suggest that the interplay between E-cadherin, YAP and glycolysis is a general biological connection. Did the authors test the role of E-cadherin expression on YAP activity and glycolysis in proliferative cells? I suggest to overexpress E-cadherin and study if it can reduce YAP activity and glycolytic flux rate;

- The cytoplasmic or mitochondrial NAD⁺/NADH ratio is one major determinant of cellular redox state, and intracellular NADPH levels are correlated with intracellular Radical Oxygen Species (ROS) and contribute to their detoxification. When ROS levels are elevated, the concentration of the oxidized form of NAD⁺ should be even higher to buffer an excess of ROS. I suggest to the authors to measure the mitochondrial and cytoplasmic ROS levels through the use of Mitoxox and Dihydroethidium (DHE) assays, respectively.

- None of the genetic or metabolic manipulations used by the authors in this manuscript was able to rescue the quiescent state. This may depend on the fact that 100% confluent cells are quiescent by definition because they cannot proliferate any longer in this plating condition. If confluent/quiescent cells are replated in lower confluency, which is the time required for observing cells restart their proliferation? If the authors are unable to use a different model of cell quiescence, I suggest to take confluent/quiescent cells (e.g., 10 days after initial plating), to split them in fresh medium, and to measure the time of proliferation recover in untreated cells, or in cells with E-cadherin/TEAD KO, or in cells with the downregulation of genes that were found to be upregulated in quiescent cells by GSEA. This analysis may reveal potential activators of quiescence inducers that could also apply to different contexts or quiescent models.

- The authors should clarify the potential consequences of their discoveries in the understanding of the biology, physiology and pathology of fibroblasts and connective tissues in humans.

- Did the authors evaluate quiescence-induced modifications of intracellular or extracellular amino acids?

Reviewer #4 (Remarks to the Author):

This is an interesting study in which the authors provide evidence that mouse fibroblasts undergo a metabolic shift when switching from a proliferative to quiescent state. In particular, quiescent (Q) fibroblasts exhibit decreased glycolysis and increased TCA cycle/OXPHOS activity. Elevated mitochondrial metabolism in Q cells is required to generate ATP and amino acids to maintain synthesis and secretion of ECM proteins. The authors identify a novel regulatory pathway in which elevated E-cadherin in Q cells triggers suppression of the transcription factor YAP1 and subsequent downregulation of glycolytic gene expression. Further evidence is presented that elevated

mitochondrial pyruvate carrier (MPC) expression drives increased flux of pyruvate into the mitochondrial TCA cycle. Additional mechanisms were also identified related to enzymes controlling the malate-to-pyruvate and oxaloacetate-to-malate flux. Quiescence in most cell types is typically associated with a reduction in basal metabolic activity, ATP production and biosynthesis. Other studies have also shown that fibroblasts appear to be the exception and increase OXPHOS to support synthesis of ECM proteins. This study provides novel insight into some of the regulatory mechanisms controlling this unusual metabolic switch in quiescent fibroblasts. However, some issues need to be addressed as follows:

1. The authors employed a mouse embryonic fibroblast cell culture model immortalized with a dominant-negative p53. The cells are triggered to undergo quiescence following 6 days of contact inhibition. How physiologically representative is this model compared to quiescent fibroblasts in adult tissues? How translatable are these findings to human fibroblasts? There is some data presented on human BJ fibroblasts (Sup Fig S3) but it is not specified how these fibroblasts were induced into quiescence

2. In Figure 1B there are still some Q cells in S and G2/M phase. What fraction of the Q cells are in G1, S and G2/M?

3. The authors fail to take into consideration changes in mitochondrial number/density between P and Q cells. It is possible that Q cells actually increase mitochondrial biogenesis which could contribute to increased TCA cycle activity and OXPHOS.

4. The connection between E-cadherin and Yap1 is not clearly explained.

5. Deletion of E-cadherin or overexpression of constitutive YAP1 clearly results in elevated glycolysis, but also results in further elevated TCA cycle activity and OXPHOS in Q cells, presumably in a YAP1 independent manner. The authors should speculate on some potential mechanism(s) for this phenomenon in the discussion.

Minor

1. The methods section should provide details on how quiescence was induced in the MEFS, NIH 3T3 and Human BJ fibroblasts.

2. It is not specified how cells were counted to assess proliferation rates.

3. It would be helpful to label key enzymes (i.e. MDH1, ME1, LDH, etc) on metabolic pathways in Fig 1 D

Response to Reviewers' Comments

We thank the reviewers for their thoughtful comments and suggestions. To address the reviewers' comments, we included 22 new figure panels in the revised manuscript. We addressed the reviewers' comments point-by-point below:

Reviewer #1 (Remarks to the Author):

Overall, the manuscript reports valuable information concerning this well characterized model in which proliferating fibroblasts transitions between serum-induced proliferative function and serum-dependent effector function (also called senescence by some). The data will be of use to others involved in this well-studied cell model. Before publication, the following issues should be addressed:

1. The authors should provide data concerning the transcriptional and metabolic differences between their wildtype MEFs and the MEFs carrying a p53 dominant-negative used in these studies. p53 has been established as a key transcriptional regulator of metabolic genes and how much of the Yap-independent mechanisms reported in the manuscript can be accounted for by p53-dependent changes in transcription should be determined since these reagents are on hand.

Response: We agree with the reviewer that p53 could affect the transcription of metabolic genes. However, the contact inhibition mediated quiescence is mainly determined by p27 and not by the p53-p21 axis. Nevertheless, we found that the major metabolic reprogramming events occurring in the immortalized MEFs are also occurring in the primary WT MEFs. We show that E-cadherin is also induced in WT MEFs as well as YAP translocation and CTGF (Fig. S5a and Fig. S5j of the revised manuscript). We also showed that glycolysis is reduced, and oxidative phosphorylation is elevated in WT MEFs, the same as in the DN-p53 MEFs by Seahorse assay (Fig. S4c and Fig. S4d). Finally, we showed in the revised manuscript the same regulation of major metabolic enzymes in DN-p53 immortalized MEFs and WT MEFs (Fig. S4a and S4b in the revised manuscript). Glycolytic enzyme levels consistently expressed at lower level in Q (quiescent) compared to P (proliferating) in WT MEFs. Conversely, MPC1-2 and several TCA cycle enzyme levels were consistently higher in Q than P WT MEFs. These findings strongly suggest that the observed metabolic shifts between proliferating and quiescent cells are not dependent on p53, implying the presence of an independent metabolic reprogramming mechanism. We used DN-p53 immortalized cells because it is impossible to genetically manipulate the primary cells and to repeat the experiments multiple times.

2) The introduction and discussion should be revised to make it clear that the data reported apply to only one well-characterized model of embryonic fibroblasts quiescence, that of contact-inhibition. The other major model of embryonic fibroblasts quiescence that is also widely studied is that of serum-deprivation. While many past studies have utilized both models to study non-proliferating fibroblasts, there are clear differences between phenotypes of fibroblasts that have ceased proliferating following serum starvation and those that arrest through contact-inhibition in the presence of serum growth factors. This should be clarified for the general reader.

Response: We modified the introduction accordingly, but in the manuscript, we discussed a recent paper showing that in quiescent neural stem cells, OXPHOS is elevated with concomitant induction of MPC1/MPC2 expression (PMID: 36857455). Thus, at least some of the metabolic changes that we see in quiescent cells are not restricted to fibroblasts. In the revised manuscript

we show that similar changes are occurring in Chinese hamster ovary (CHO) cells, derived from epithelial cells of the ovary of the Chinese hamster, namely, reduced glycolytic enzymes and ECAR and upregulated TCA cycle enzymes and OCR in Q cells (Fig. S4e-h in the revised manuscript). Finally, in the revised manuscript we show that quiescent MEFs after serum starvation reduced both glycolysis and OXPHOS (Fig. S11), suggesting that growth factors in contact inhibited Q cells are required for the elevation of TCA cycle and OXPHOS.

Reviewer #2 (Remarks to the Author):

Kang et al. use metabolic flux analysis in mouse embryonic fibroblasts (MEFs) to investigate the metabolic differences of proliferating and quiescent cells. They find extensive metabolic rewiring during the transition from the proliferative towards the quiescent state, most prominently a reduction in glycolytic flux with a concomitant increase in TCA cycle flux in quiescent cells. The authors went on to show that increased E-Cadherin expression in quiescent cells suppresses glycolytic flux via inhibition of Yap/TAZ activity. Mediators of increased TCA cycle flux may involve differential expression of the mitochondrial pyruvate carrier. Finally, the authors propose that the metabolic rewiring observed in quiescent fibroblasts supports extracellular matrix synthesis.

The authors provide a fresh perspective on how cellular metabolism supports the metabolic requirements of proliferative and non-proliferative cells and thereby address a gap in the field. Several of their findings are novel and should attract interest in the field, including the inhibition of glycolysis via E-Cadherin mediated suppression of YAP/TAZ, and the impaired dimerization of ME1, in the quiescent cells. The metabolic experiments are technically sound and support most of their conclusions. However, there are several major issues that I list below and suggest the authors to address before consideration for publication. While some mechanistic studies on the underpinnings of the metabolic phenotypes have been performed, they are incomplete. In addition, some important literature is not cited. Overall, the study falls short of connecting the observed metabolic differences to any functional consequences, which reduces the significance of the reported findings. Lastly, the relevance of the findings beyond fibroblasts remains unclear.

Response: In the case of fibroblasts, we showed in the revised manuscript that in Q cells the production of collagens in the ECM is dependent on the increase of MPCs and OXPHOS (Fig.8 h-i in the revised manuscript). With respect to other cells, we cited a recent paper showing that in quiescent neural stem cells MPC1 and MPC2 are elevated with concomitant elevation of OXPHOS (PMID: 36857455). In the revised manuscript we showed that this is also true in quiescent CHO cells (Fig. S4f in the revised manuscript).

1. The authors need to provide more details on the ¹³C-MFA experiment in the methods as this is the key experiment of the study. For example, they state that the cells were plated and pulse-labeled the next day, which indicates they haven't been subjected to the same proliferation/quiescence protocol as described in Figure 1A. Have they been treated like this and then replated for the tracer studies? In addition, it remains unclear which of the time points for the tracing described was used to derive the model. The authors should also describe how they verified the nature of the metabolites measured.

Response: We thank the reviewer for this comment. Indeed, mistakenly we wrote in the Methods section that a day after plating the cells were labelled. In fact, the medium was replaced with medium containing either 25 mM [1,2-¹³C] glucose or 4 mM [U-¹³C] glutamine when the cells were actively proliferating (2 days after plating) and the cells were quiescent (contact-inhibited for 6 days, 11 days after plating) as described in Fig. 1 A. Samples were collected 6, 18, and 24 h after tracer addition. We have corrected this in the revised manuscript. We updated the materials and methods section with more details regarding how ¹³C-MFA was performed, what measurements were collected, how metabolites were verified, and what analytical approaches were used.

2. A similar problem exists for the extracellular flux analysis - how were the cells brought into the P and Q states for seahorse assays?

Response: As described in Figure 1a, for the tracer studies, the cells were analyzed while actively proliferating (2 days after plating) and when they were quiescent (contact-inhibited for 6 days, 11 days after plating). After reaching proper growth states (P and Q), cells were replated at 1.5×10^3 cells per well in Seahorse Bioscience 96-well plate. After settling, the extracellular acidification rate (ECAR) and oxygen consumption rate (OCR) were measured using an Agilent Seahorse XFe96 Analyzer using standard methods.

3. The authors suggest higher MPC expression promotes the increased pyruvate flux into mitochondria. How is MPC expression regulated in quiescent vs proliferative cells?

Response: To find the exact mechanisms by which MPCs are regulated is a major project beyond the scope of these studies. We would like to point out that a recent manuscript showed that MPCs are also elevated in quiescent neural stem cells (PMID: 36857455). In the revised manuscript we showed that MPC1/MPC2 are also elevated in quiescent CHO cells, which are epithelial cells (Fig. S4h). Finally, in the revised manuscript we show that overexpression of MPC1/MPC2 in proliferating cells elevates oxidative phosphorylation (Fig. S3f), whereas inhibition of MPC decreases oxidative phosphorylation in quiescent cells (Fig. 1m).

4. Some of the transcriptomic changes observed by the authors in the quiescent cells seem to be specific to fibroblasts, including those related to the ECM. To broaden the significance of their study, the authors should perform some of their key experiments in other relevant cell types, comparing the proliferative and quiescent state.

Response: In the revised manuscript, we show that similar metabolic changes and expression of glycolytic and TCA cycle enzyme are occurring in Chinese Hamster Ovary (CHO) cells, which do not express ECM proteins (Fig. S4e-h). We repeated the major experiments with CHO cells which are epithelial cells, such as measuring ECAR for glycolysis, OCR for mitochondrial respiration by Seahorse and both ECAR and OCR show the same pattern as with the MEFs. The glycolytic enzymes levels are consistently lower in Q than P CHO cells. MPC1-2 and TCA cycle enzymes level are consistently higher in Q than P CHO cell. These data suggest that the metabolic reprogramming observed between proliferating and quiescent cells is a more common phenomenon. As extracellular matrix (ECM) production is specific to fibroblasts, the expression of ECM was not observed in CHO cells.

5. The authors use AOA as a MAS inhibitor. However, AOA broadly inhibits many transaminases and is not specific to the MAS. Therefore, to support their conclusions, the authors need to provide more evidence, for example using a genetic approach. This may include deletion of Slc25a11 which is relevant in their experimental conditions as it is downregulated in Q compared to P cells.

Response: AOA is commonly used as MAS inhibitor. However, in the revised manuscript we recapitulated the results using Slc25a11 deletion as suggested by the reviewer (Fig. 7d,e and i in the revised manuscript).

6. The authors fail to cite important primary literature regarding the role of proline metabolism for collagen synthesis in fibroblasts (PMID: 30973753, 32134147, 35760868). Moreover, PYCR1 is not the key mitochondrial enzyme responsible for proline biosynthesis, but P5CS / Aldh18a1 is the rate-limiting enzyme, as shown in embryonic fibroblasts like those used in this study (PMID: 32134147).

Response: In the revised manuscript we show that both PYCR1 and P5CS are significantly upregulated in the quiescent cells, but P5CS is only modestly induced (Fig. 8g in the revised manuscript). The revised text was corrected accordingly and includes the suggested references.

7. The title of the last section needs to be corrected to “Active mitochondrial metabolism may support ECM biosynthesis in quiescent cells”, as no experimental evidence supporting this conclusion is provided.

Response: We appreciate this comment by the reviewer. In the revised manuscript, we provided the evidence for the conclusion (see below).

8. To support the significance of their findings, the authors should provide mechanistic evidence that the metabolic rewiring in the quiescent cells is functionally relevant. Are some of the observed changes required to maintain the quiescent state? Does the metabolic rewiring indeed support ECM production as suggested by the authors? Does the metabolic rewiring indeed support ECM production as suggested by the authors?

Response: MPC inhibition reduces both basal respiration and ATP production from mitochondrial respiration (Fig. 1m). In the revised manuscript we showed that inhibition of MPC or complex I respectively reduces the expression of collagens (Fig. 8h-i in the revised manuscript). These data suggest that the upregulated mitochondrial metabolism supports ECM production in quiescent cells.

9. It would also be helpful if the authors could provide a mechanism by which the increased MPC expression or higher flux of glutamine into proline is regulated in the quiescent cells.

Response: It is not clear what this comment entails. As we indicated earlier, to find the mechanism by which MPCs are upregulated is a long-term project and is beyond the objectives of this manuscript.

Minor:

1. The authors should cut down on their introduction - the end is repetitive with the discussion.

Response: We appreciate this comment by the reviewer. We updated the introduction in the revised manuscript.

2. Page 5, the authors state that “most TCA cycle enzymes ... were significantly upregulated in Q cells”, but they only look at four, which is not most.

Response: In the revised manuscript we show that 5 TCA cycle enzymes are elevated (Fig. 1I). Nevertheless, we changed the text to “some TCA cycle enzymes were significantly elevated”.

3. Page 6, end of first paragraph, the authors suggest that “attenuated glycolysis and induced mitochondrial metabolism are global phenomena in quiescent fibroblasts”, but they only ever look at metabolic enzyme expression in the other fibroblasts investigated, which is not the same as glycolysis or mitochondrial metabolism.

Response: In the revised manuscript we showed that glycolysis is downregulated and OXPHOS is elevated in quiescent NIH3T3 cells, BJ cells and in CHO cells which are epithelial cells (Fig. S4c-f). We also changed the text to “...are common...” instead of “...are global...”

4. Page 6, the authors suggest that Q E-Cad KO cells are more energetic, but no experimental data to support a higher bioenergetic state are provided.

Response: Since Q E-Cad KO cells have both higher glycolysis and OXPHOS compared to control Q cells, they are more energetic.

5. Page 7, the authors need to show successful deletion and re-expression of wt and mutant YAP in their cells, e.g. by western blot.

Response: We appreciate this comment. As shown in Fig. S7a of the revised manuscript, YAP was knocked out by CRISPR-Cas9 system and then were infected with either pBabe EV, pBabe YAP WT, or pBabe YAP 5SA to express YAP WT or YAP 5SA mutant. EV cells do not show YAP1 expression.

6. Page 9, based on Fig. 6E the authors state that ACAT1-mediated acetylation of ME1 was higher in P cells than in Q cells, but the experiment only shows a correlation between acetylation and ACAT1, not that ACAT1 mediates the acetylation of ME1 under their experimental conditions.

Response: We agree with this comment as we found only a correlation between the two. We revised the statement in the manuscript to “acetylation of ME1 is higher in P cells than in Q cells, likely mediated by ACAT1....”.

7. Page 10, the authors state that the only source of NADPH in Q cells is the PPP, but they do not show that. For the cytosolic part, they do not consider IDH1. In the statement as is, it also is not specific to the cytosol, which is what I believe the authors are trying to say. Moreover, if the Q cells don't require high NADPH as suggested by the authors, shouldn't they be less sensitive to inhibition of the PPP?

Response: IDH1 generates NADPH when is converting citrate to AKG. However, our flux analysis showed that the net flux is occurring in the opposite direction (from AKG to citrate) which consumes NADPH and produces NADP⁺ (Fig. 1e). Thus, it is unlikely that IDH1 plays a role in

generating NADPH in Q cells. Although Q cells may not require high NADPH level, cutting down the predominant source of NADPH (PPP) could increase cell death. This was previously shown in human fibroblasts (PMID: 21049082).

Reviewer #3 (Remarks to the Author):

In this manuscript, Kang S and colleagues investigate the metabolic modifications occurring during the transition from a proliferative to a quiescent state in mouse embryonic fibroblasts. They combine metabolic flux and transcriptomic analysis to identify the metabolic pathways that are the mostly modulated upon cellular switch from proliferation to quiescence. The authors nicely demonstrate that quiescent cells undergo a reduction in glucose uptake and glycolytic flux, which is mostly mediated by a reduction in the expression of glycolytic genes when compared to proliferative cells. This quiescence-induced reduction of glycolysis is largely dependent on increased E cadherin expression and consequently reduced YAP activity. While glycolysis is inhibited, the flux of pyruvate into mitochondria and the oxidation acetyl-CoA in the TCA cycle and OXPHOS pathways are markedly increased in quiescent cells in YAP-independent manner, and an increased expression of MDH1 supports leading to a global metabolic rewiring that stimulates the conversion of cytosolic malate to oxaloacetate, PEP and pyruvate.

The study is rigorously conducted and provides a global view of the main metabolic changes occurring in quiescent immortalized fibroblasts when compared to proliferating ones. While confirming previously published data (i.e., that non proliferating cells switch from glycolysis to mitochondrial oxidation of pyruvate and acetyl-CoA), this study reveals new details about the general rewiring of crucial metabolic pathways, as well as of NAD⁺ and NAD⁺/NADH ratio, in quiescent cells when compared to proliferating ones.

The manuscript is written in a very clear way. The experimental approach is also appropriate, and individual experiments are interpreted in the correct way.

Therefore, in my opinion this manuscript could be considered for publication in Nature Communications.

However, I have major concerns that must be addressed before further considering this work for publication.

Major issues

- The main limitation of this study is the fact that the quiescent state of fibroblasts is achieved by allowing cells reach confluence in the growth plate. Therefore, it is difficult to understand what metabolic changes occurring in quiescent cells are the results of quiescence itself, and what metabolic adaptive modifications are the result of the fact that cells reach 100% confluence in the plate, which comes with a significant reduction of glucose and amino acid concentration in cell growth media. For instance, while an increased expression of E cadherin is clearly responsible for reduced YAP expression and lowered glycolytic flux in quiescent cells, it is unclear whether the observed increased in mitochondrial oxidation of pyruvate and acetyl-CoA depends on E-cadherin-induced quiescence or, alternatively, on cellular confluence and consequent shortage of extracellular glucose and other metabolites. Similarly, it is unclear whether an increased expression of mitochondrial transporters of pyruvate is essential for the survival of quiescent cells because of changes in their proliferative state, or because in the experimental system used in this

work quiescence reflects a drop in extracellular metabolites, which forces tumor cells to enhance mitochondrial pyruvate oxidation to survive. To address this point, I suggest to replicate the major study findings by also using an alternative experimental approach to induce cellular quiescence.

-... it is difficult to understand what metabolic changes occurring in quiescent cells are the results of quiescence itself, and what metabolic adaptive modifications are the result of the fact that cells reach 100% confluence in the plate, which comes with a significant reduction of glucose and amino acid concentration in cell growth media.

Response: The quiescence in our system is achieved by cell-cell interaction (contact inhibition). In this case E-cadherin is activated in one cell by interaction with E-cadherin in another cell. As indicated below the contact inhibited cells are not limited for growth factors.

-...it is unclear whether the observed increase in mitochondrial oxidation of pyruvate and acetyl-CoA depends on E-cadherin-induced quiescence or, alternatively, on cellular confluence and consequent shortage of extracellular glucose and other metabolites.

Response: As shown in Figure S4d-e, although E-Cadherin mediates the reduced glycolysis in Q cells but not the increase in the TCA cycle and OXPHOS. The increased mitochondrial oxidation of pyruvate and acetyl-CoA are not dependent on E-cadherin-induced quiescence because quiescent E-cad KO cells still maintained high OXPHOS. As the cell culture medium was changed every other day (48h), extracellular metabolites were sufficient. Also, as MPC1 and MPC2, and TCA cycle enzymes (IDH2 and SDHA) levels, were maintained at high levels in the Q state of E-Cad KO, TEAD KO, and YAP 5SA cells (Fig. S6i-k), mitochondrial oxidation of pyruvate and acetyl-CoA does not depend on E-cadherin.

I suggest replicating the major study findings by also using an alternative experimental approach to induce cellular quiescence.

Response: We appreciate this comment. We did not use serum deprivation because it is governed by the effect of growth factors. Nevertheless, in the revised manuscript, we show that MEFs, which were made quiescent by serum deprivation showed both reduced glycolysis and OXPHOS (Fig. S11 of the revised manuscript), suggesting that the presence of growth factors in the contact inhibited Q cells is required for the induction of OXPHOS.

- How many times did the authors change the growth medium in quiescence cells? The authors state that the quiescent state is reached because of contact-mediated inhibition of fibroblast proliferation. However, fibroblasts may also stop proliferating because of the shortage of extracellular metabolites, such as glucose or amino acids, in the growth medium of confluent cells after several days of growth. To circumvent this major limitation of the experimental system used in this work, I suggest to repeat at least the major experiments by refreshing medium in order to exclude the possibility that the reduced glycolytic flux should be dependent on the absence of glucose in the growth medium 11 days after seeding.

Response: We apologize for the confusion. We changed the medium every other day for both P and Q cells and all assays (western blot, tracing, RT-qPCR) were done 24h after changing medium unless it is indicated. The remaining glucose concentration was sufficient with both P and Q cells after 24 hours of culture. The Methods section was revised accordingly.

- I disagree with one of the main study conclusions, i.e., that the metabolic switch occurring during the transition into a quiescent state is a potentially universal mechanism that may occur in different

cell types, including normal epithelial cells or even cancer cells. Indeed, different cell types can be highly heterogeneous in terms of metabolic requirements and metabolic responses to external stress or changes in their microenvironment. In this work, the authors only used a single cell line, which was represented mouse embryonic fibroblasts (MEFs), and they just confirmed that some of the TCA cycle enzymes are increased on another murine quiescent cell line, NIH 3T3, and in human BJ fibroblasts. I recommend to replicate the main study findings in at least another model of non-tumoral cells, and in at least one model of cancer cells.

Response: In the revised manuscript we showed that in Q NIH3T3 and BJ cells glycolysis (ECAR) is decreased and OXPHOS (OCR) is elevated (Fig. S4c-d in the revised manuscript). In the revised manuscript we also repeated the experiment with Chinese hamster ovary (CHO) cells, derived from epithelial cells of the ovary of the Chinese hamster. We found reduced glycolytic enzymes and ECAR and upregulated TCA cycle enzymes and OCR in Q cells (Fig. S4e-h in the revised manuscript). We did not repeat the experiment with cancer cells because cancer cells don't go into quiescence by contact inhibition.

- Why did the authors not evaluate the expression of a lot of glycolytic enzymes which were studied in the case of MEFs (Fig 1 K)? Why did they just evaluate the expression of HK2 and PFK1 in the case of NIH 3T3 and BJ cells? I suggest to show the expression of the other glycolytic enzymes in the two additional models;

Response: In the revised manuscript we examined more enzymes (Fig. S4a-b). Similar to MEFs, most rate-limiting glycolytic enzymes are downregulated. However, unlike in MEFs not all glycolytic enzymes are downregulated.

- What is the impact of YAP overexpression on quiescence-induced metabolic rewiring?

Response: As we showed in Fig. 4f, constitutively active YAP in Q cells further enhanced both glycolysis and the TCA cycle fluxes. Consistently, the basal respiration measured by the OCR was much higher in Q 5SA cells than in P cells (Fig. S7h).

- Which is the correlation between YAP constitutively activation and an increment in TCA cycle in quiescent cells? I suggest to demonstrate with ^{13}C glucose flux that that an increased glycolysis, caused by YAP activation, could in turn induce an even higher activation of TCA cycle;

Response: Our flux analysis using ^{13}C glucose and glutamine (Fig. 4f) in cells expressing activated YAP clearly show that the increase in glycolysis increases the flux of pyruvate to the mitochondria and the TCA cycle.

- Since it is well known from the literature that YAP is implicated in cell migration, I suggest to the authors to evaluate the impact of YAP overexpression or KO on MEF cell migration by using a wound healing assay in non-confluent (non-quiescent) cells.

Response: We think that addressing this comment is a diversion from the main topic of the manuscript.

- I disagree with the interpretation of the authors about the reason why ME1 flux is suppressed in quiescent cells. Indeed, in order to prove that dimerization of ME1, and consequently also its activity, is reduced because of an activation of PPP, which require to quiescent cells the proper amount of NADPH, a KO of some genes involved in the PPP pathway is required. If the hypothesis

is correct, ME1 dimerization should be observed. I suggest to KO some of the limiting enzyme of PPP pathway, like Glucose-6-phosphate dehydrogenase.

Response: We apologize for the misunderstanding. Our conclusion is based on the flux analysis, which showed that the flux mediated by ME1 was diminished, and by the Co-IP results which showed a decrease in ME1 dimerization. The PPP is not increased in Q cells but is rather maintained and therefore there is no reason to believe that the reduction of ME1-mediated flux is due to the increase in PPP. As discussed in the Discussion section, we propose that this phenomenon may be influenced by the relatively higher levels of cytosolic NAD⁺ in Q cells, which could activate SIRT6-mediated ME1 deacetylation and subsequent deactivation.

- The authors clarify very well the role of YAP in metabolic reprogramming by reducing glycolytic flux in MEFs quiescent cells. Anyway, they did not rescue the quiescent state of cells with constitutively active form of YAP, which let them to conclude that quiescent state of cells is independent on YAP. Have you considered the idea to use a different type of model? If the model is characterized by cells which completely covered the plate and achieve 100% confluence, how did the authors can expect a rescue of quiescence? I suggest to repeat the experiment by inducing cells quiescence through serum starvation;

Response: We did not intend to rescue the quiescence by expressing activated YAP. Quiescence is the cause of metabolic reprogramming, which includes reduced glycolysis and elevated OXPHOS, and not vice versa. In the revised manuscript, we induced quiescence by serum starvation. However, this quiescence is governed by the lack of growth factors. We found that the lack of growth factors decreased both glycolysis and OXPHOS (Fig. S11) (see also response to comment above).

- Did the authors have check the expression of senescence markers, such as p21 or b-galactosidase, in quiescent cells? ;

Response: The quiescent cells do not senesce because quiescence is reversible exit of cell cycle and if the quiescent cells are sparsely replated and are not contact inhibited, they proliferate again (see below a figure for the reviewer).

- The authors indirectly suggest that the interplay between E-cadherin, YAP and glycolysis is a general biological connection. Did the authors test the role of E-cadherin expression on YAP activity and glycolysis in proliferative cells? I suggest to overexpress E-cadherin and study if it can reduce YAP activity and glycolytic flux rate;

Response: We clearly show that deletion of E-cadherin in Q cells is suppressing YAP activity. In order for E-Cadherin to affect YAP, E-Cadherin of one cell should interact with E-cadherin of other cells, which occurs only when the cells contact each other. Expressing YAP in proliferating cells may not exert the same effect because the cells are not contact inhibited.

- The cytoplasmic or mitochondrial NAD⁺/NADH ratio is one major determinant of cellular redox state, and intracellular NADPH levels are correlated with intracellular Radical Oxygen Species (ROS) and contribute to their detoxification. When ROS levels are elevated, the concentration of the oxidized form of NAD⁺ should be even higher to buffer an excess of ROS. I suggest to the authors to measure the mitochondrial and cytoplasmic ROS levels through the use of Mitoxox and Dihydroethidium (DHE) assays, respectively.

Response: NADPH is important for scavenging ROS. The mitochondrial NAD⁺/NADH ratio could determine superoxide production in the mitochondria. However, it is not clear to us why the reviewer wanted us to measure ROS and how it is related to the results in the manuscript. Additionally, ROS levels can be regulated by multiple mechanisms, in addition to NADPH and NADH levels, such as the level of anti-oxidants MnSOD and catalase.

- None of the genetic or metabolic manipulations used by the authors in this manuscript was able to rescue the quiescent state. This may depend on the fact that 100% confluent cells are quiescent by definition because they cannot proliferate any longer in this plating condition. If confluent/quiescent cells are replated in lower confluency, which is the time required for observing cells restart their proliferation?

Response: Quiescence is a reversible exit of cell cycle. When we replated the Q cells (Cl6; contact inhibited for 6 days) in lower confluency, they immediately re-proliferated and showed P cells' phenotypes, such as reduced Collagen, p27^{Kip1}, and induced HK2 and PKM1 levels. (See also response to previous comment above).

- Did the authors evaluate quiescence-induced modifications of intracellular or extracellular amino acids?

Response: As we describe in the updated methods section, we quantified both changes in extracellular uptake and secretion rates, which included rates of amino acid uptake and secretion, as well as changes in intracellular metabolic fluxes. The details can be found in Table S3.

Reviewer #4 (Remarks to the Author):

This is an interesting study in which the authors provide evidence that mouse fibroblasts undergo a metabolic shift when switching from a proliferative to quiescent state. In particular, quiescent (Q) fibroblasts exhibit decreased glycolysis and increased TCA cycle/OXPHOS activity. Elevated mitochondrial metabolism in Q cells is required to generate ATP and amino acids to maintain synthesis and secretion of ECM proteins. The authors identify a novel regulatory pathway in which elevated E-cadherin in Q cells triggers suppression of the transcription factor YAP1 and subsequent downregulation of glycolytic gene expression. Further evidence is presented that elevated mitochondrial pyruvate carrier (MPC) expression drives increased flux of pyruvate into the mitochondrial TCA cycle. Additional mechanisms were also identified related to enzymes controlling the malate-to-pyruvate and oxaloacetate-to-malate flux. Quiescence in most cell types is typically associated with a reduction in basal metabolic activity, ATP production and biosynthesis. Other studies have also shown that fibroblasts appear to be the exception and increase OXPHOS to support synthesis of ECM proteins. This study provides novel insight into some of the regulatory mechanisms controlling this unusual metabolic switch in quiescent fibroblasts. However, some issues need to be addressed as follows:

1. The authors employed a mouse embryonic fibroblast cell culture model immortalized with a dominant-negative p53. The cells are triggered to undergo quiescence following 6 days of contact inhibition. How physiologically representative is this model compared to quiescent fibroblasts in adult tissues?

Response: This is an important question. As described in the Introduction, fibroblasts are the most common cell types in mammalian connective tissue. While these cells proliferate during the embryonic stage, they typically enter a state of G1 cell cycle arrest with age. Although healthy fibroblasts in normal tissue are mostly in a quiescent state, they re-enter the cell cycle under specific conditions, such as in response to wound healing. Our study provides insights into the metabolic reprogramming processes and their regulatory mechanisms during this transition. It seems that elevated OXPHOS is a common phenomenon in contact inhibited quiescent cells. This is also happening in human BJ cells, which are skin fibroblasts (Fig. S4d). It was recently shown that in quiescent neural stem cells, OXPHOS is elevated with concomitant induction of MPC1/MPC2 expression (PMID: 36857455). In the revised manuscript we showed that same phenomena occur in CHO cells (Fig, S4f).

How translatable are these findings to human fibroblasts? There is some data presented on human BJ fibroblasts (Sup Fig S3), but it is not specified how these fibroblasts were induced into quiescence

Response: Same as described in Figure 1A, human BJ fibroblasts were analyzed while actively proliferating (2 days after plating) and when they were quiescent (contact-inhibited for 6 days, 11 days after plating). After reaching proper growth states (P and Q), the enzyme expression levels were assessed. Culture medium was changed every other day in both P and Q cells. The Method section was revised accordingly.

2. In Figure 1B there are still some Q cells in S and G2/M phase. What fraction of the Q cells are in G1, S and G2/M?

Response: As indicated better in (Fig. S6b, see Q sgCdh1 (-) control), Q cells were arrested in the G1 cell cycle phase. The mean fraction of the Q cells in G1 is 96.50%, S is 1.53%, and G2/M is 1.99% respectively.

3. The authors fail to take into consideration changes in mitochondrial number/density between P and Q cells. It is possible that Q cells actually increase mitochondrial biogenesis which could contribute to increased TCA cycle activity and OXPHOS.

Response: Morphometric analysis of mitochondria revealed that Q cells contained primarily fused and elongated mitochondria, whereas P cells had more fragmented mitochondria (Fig. S3b-c in the revised manuscript). Mitochondrial membrane potential representing mitochondrial activity quantified by TMRE is higher in Q cells than P cells (Fig. 1g-h). Together, these findings show that Q cells have an elongated mitochondrial network and higher mitochondrial activity. Interestingly the same observation was reported for quiescent neural stem cells (PMID: 36857455).

4. The connection between E-cadherin and Yap1 is not clearly explained.

Response: As we indicated in the text of the manuscript, E-cadherin was previously reported to activate the Hippo pathway and to inhibit Yap activity. In the manuscript we showed that the deletion of E-cadherin increased YAP nuclear localization and activity (Fig. 3).

5. Deletion of E-cadherin or overexpression of constitutive YAP1 clearly results in elevated glycolysis, but also results in further elevated TCA cycle activity and OXPHOS in Q cells, presumably in a YAP1 independent manner. The authors should speculate on some potential mechanism(s) for this phenomenon in the discussion.

Response: As we indicated in the text, the increase in the glycolytic flux by the constitutively active YAP further increase the level of pyruvate available for the TCA cycle.

Minor

1. The methods section should provide details on how quiescence was induced in the MEFS, NIH 3T3 and Human BJ fibroblasts.

Response : Same as described in Figure 1A, NIH 3T3 and human BJ fibroblasts were analyzed while actively proliferating (2 days after plating) and when they were quiescent (contact-inhibited for 6 days, 11 days after plating). After reaching proper growth states (P and Q), the enzyme expression levels were assessed. Culture medium was changed every other day in both P and Q cells. The Method section was revised accordingly.

2. It is not specified how cells were counted to assess proliferation rates.

Response: Cells were counted using a mixed solution of propidium iodide (PI) and Hoechst 33342 (Hoechst) with a final concentration of 2 µg/mL PI and 10 µg/ml Hoechst for 30 min at 37 °C. Plates were then imaged on a Celigo Image Cytometer (Nexcelcom Bioscience), and the number of PI-positive cells was divided by the total number of cells (Hoechst positive) to show the percentage of dead cells (% Dead). The following formula was used to calculate the proliferation rate: Doublings per day = $[\log_2(\text{final day 2 cell count}/\text{initial day 0 cell count})]/2 \text{ days}$. The Method section was revised accordingly.

3. It would be helpful to label key enzymes (i.e. MDH1, ME1, LDH, etc) on metabolic pathways in Fig 1 D

Response: The enzymes are labeled in Fig. 1d.

REVIEWERS' COMMENTS

Reviewer #1 (Remarks to the Author):

The manuscript has been revised and additional experiments included to address the issues raised in the prior review as outlined in the author's response.

Reviewer #2 (Remarks to the Author):

In their revised manuscript, the authors have addressed most of my initial concerns. However, two concerns remain which relate to the experimental system used and the generalizability of the findings.

I agree with the other reviewers that the observed metabolic changes should be confirmed in another model of quiescence if the authors want to generalize their findings. The new CHO cell data were generated with the same model, it therefore is unclear if the metabolic changes observed are a general feature of quiescent cells or of contact inhibition only. The others state that they didn't repeat the experiments in cancer cells as they do not undergo contact inhibition, supporting the idea that the findings may only be relevant in this context. Referring to findings in another study to make does not necessarily support their claim of generalizability. This is a significant shortage in this manuscript and has not been sufficiently addressed in the revised manuscript, including the serum deprivation experiments which has the caveat of reduced growth factor availability. This dampens the significance of the findings.

In addition, the authors did not address the point raised by me and another reviewer regarding the requirement of the observed metabolic changes in maintaining the quiescent state. While new data provided support the idea that these metabolic changes support ECM synthesis which is upregulated in the Q cells, it remains unclear how these metabolic changes relate to the quiescent state itself. As reviewer 3 indicated, this may be an artefact of the contact inhibition system which maintains the quiescence, and as such, perturbing the metabolic changes in this system might not allow the authors to address this important question. This further supports the need for an alternative experimental model of quiescence by the authors as suggested above.

I therefore ask the authors to perform additional studies to support that their findings are conserved in other models of quiescence.

Reviewer #3 (Remarks to the Author):

Most of my points and critiques have been convincingly addressed.

The authors made quite a good work in reviewing the manuscript, and in particular in conducting additional experiments, adding new data and modifying the text and figures, as suggested. This has significantly improved the global quality of this work.

Therefore, the revised manuscript can be accepted for publication.

Reviewer #4 (Remarks to the Author):

The authors have addressed my concerns by conducting more experiments, revising the text and figures, and providing clarification on requested details. The manuscript is suitable for publication.

Reviewer's #2 comment: In their revised manuscript, the authors have addressed most of my initial concerns. However, two concerns remain which relate to the experimental system used and the generalizability of the findings.

I agree with the other reviewers that the observed metabolic changes should be confirmed in another model of quiescence if the authors want to generalize their findings. The new CHO cell data were generated with the same model, it therefore is unclear if the metabolic changes observed are a general feature of quiescent cells or of contact inhibition only. The others state that they didn't repeat the experiments in cancer cells as they do not undergo contact inhibition, supporting the idea that the findings may only be relevant in this context. Referring to findings in another study to make does not necessarily support their claim of generalizability. This is a significant shortage in this manuscript and has not been sufficiently addressed in the revised manuscript, including the serum deprivation experiments which has the caveat of reduced growth factor availability. This dampens the significance of the findings.

In addition, the authors did not address the point raised by me and another reviewer regarding the requirement of the observed metabolic changes in maintaining the quiescent state. While new data provided support the idea that these metabolic changes support ECM synthesis which is upregulated in the Q cells, it remains unclear how these metabolic changes relate to the quiescent state itself. As reviewer 3 indicated, this may be an artefact of the contact inhibition system which maintains the quiescence, and as such, perturbing the metabolic changes in this system might not allow the authors to address this important question. This further supports the need for an alternative experimental model of quiescence by the authors as suggested above.

I therefore ask the authors to perform additional studies to support that their findings are conserved in other models of quiescence.

Response: We appreciate the concerns by this reviewer. However, to our knowledge there are two major ways to induce quiescence; contact inhibition and serum deprivation. We conducted the serum deprivation experiment in response to reviewer's comment. To our knowledge the contact inhibition mediated quiescence is the most physiologically relevant. We are not aware of another experimental model of quiescence.